# Optimizing high-temperature energy storage in tungsten bronze-structured ceramics via high-entropy strategy and bandgap engineering

Yangfei Gao [1,7], Zizheng Song[2,7], Haichao Hu[1], Junwen Mei[1], Ruirui Kang[1], Xiaopei Zhu[3], Bian Yang [3], Jinyou Shao [1,4], Zibin Chen [2] ✉, Fei Li [5], Shujun Zhang [6] ✉ & Xiaojie Lou [1] ✉

As a vital material utilized in energy storage capacitors, dielectric ceramics have widespread applications in high-power pulse devices. However, the development of dielectric ceramics with both high energy density and efficiency at high temperatures poses a significant challenge. In this study, we employ high-entropy strategy and band gap engineering to enhance the energy storage performance in tetragonal tungsten bronze-structured dielectric ceramics. The high-entropy strategy fosters cation disorder and disrupts long-range ordering, consequently regulating relaxation behavior. Simultaneously, the reduction in grain size, elevation of conductivity activation energy, and increase in band gap collectively bolster the breakdown electric strength. This cascade effect results in outstanding energy storage performance, ultimately achieving a recoverable energy density of 8.9 J cm$^{-3}$ and an efficiency of 93% in $Ba_{0.4}Sr_{0.3}Ca_{0.3}Nb_{1.7}Ta_{0.3}O_6$ ceramics, which also exhibit superior temperature stability across a broad temperature range up to 180 °C and excellent cycling reliability up to 10$^5$. This research presents an effective method for designing tetragonal tungsten bronze dielectric ceramics with ultra-high comprehensive energy storage performance.

With the continuous progress in the electronics and automotive industries, there is a growing demand for advanced energy storage materials in electric vehicles (EVs) and power electronics, particularly for pulse power applications[1]. Dielectric ceramics, renowned for their ultra-fast discharge rates, superior power density, and excellent high-temperature resistance, have garnered considerable interest in energy storage applications. However, their practical implementation is impeded by their low recoverable energy storage density ($W_{rec}$) and low efficiency ($\eta$)[2].

The energy storage performance of a dielectric depends on its dielectric polarization ($P$) under an externally applied electric field ($E$),

[1]Frontier Institute of Science and Technology, State Key Laboratory for Mechanical Behavior of Materials, and Xi'an Key Laboratory of Electric Devices and Materials Chemistry, Xi'an Jiaotong University, Xi'an, China. [2]Department of Industrial and Systems Engineering, The Hong Kong Polytechnic University, Hong Kong, China. [3]School of Materials Science and Engineering, Xi'an University of Technology, Xi'an, Shaanxi, China. [4]Micro-and Nano-Technology Research Center, State Key Laboratory for Manufacturing Systems Engineering, Xi'an Jiaotong University, Xi'an, China. [5]Electronic Materials Research Laboratory (Key Lab of Education Ministry), State Key Laboratory for Mechanical Behavior of Materials and School of Electronic and Information Engineering, Xi'an Jiaotong University, Xi'an, China. [6]Institute for Superconducting and Electronic Materials, Faculty of Engineering and Information Sciences, University of Wollongong, Wollongong, NSW, Australia. [7]These authors contributed equally: Yangfei Gao, Zizheng Song. ✉e-mail: zi-bin.chen@polyu.edu.hk; shujun@uow.edu.au; xlou03@mail.xjtu.edu.cn

that is, $W_{total} = \int_0^{P_{max}} E dP, W_{rec} = \int_{P_r}^{P_{max}} E dP, \eta = W_{rec}/W_{total} \cdot 100\%$, where $W_{total}$, $W_{rec}$, $\eta$, $P_{max}$ and $P_r$ represent the total energy storage density, recoverable energy storage density, efficiency, maximum polarization, and remnant polarization after discharging, respectively[3]. According to the above definition, the key to achieve excellent energy storage density is to increase $P_{max}$ while reducing $P_r$ (i.e., obtaining high $\Delta P = P_{max} - P_r$) and enhancing $E_b$, the breakdown strength, which is closely associated with the maximum applied electric field the ceramics can withstand.

The tetragonal tungsten bronze structure (TTBs), regarded as the second largest family of ferroelectrics, has not attracted sufficient attention due to its complex crystal structure, low breakdown strength and suboptimal property in energy storage. TTBs derive from the perovskite structure but offer enhanced flexibility in customizing local stoichiometry and lattice structures. The unit cell of TTBs, represented by the general formula $(A1)_2(A2)_4(C)_4(B1)_2(B2)_8O_{30}$, consists of layers of corner-sharing $BO_6$ octahedra with three types of interstitial channels: quadrilateral A1, pentagon A2, and triangular C channels. TTBs dielectrics are classified into three types based on ion occupancy at lattice sites: fully filled, filled, and unfilled. The intricate crystal structure of TTBs ceramics offers various opportunities for customizing ionic composition, accommodating different charges and radii[4].

$Sr_xBa_{1-x}Nb_2O_6$ (SBN) is an actively studied composition featuring a tetragonal tungsten bronze structure (TTBs). Unlike filled structures, SBN is classified as an unfilled TTBs due to the presence of a 1/6 vacancy at the A site, which is randomly distributed in the A1 and A2 sites. Typically, the larger $Ba^{2+}$ ions exclusively occupy the A2-site, while the smaller $Sr^{2+}$ ions are distributed among both A1 and A2 sites[5]. The widely accepted explanation for the relaxation behavior in TTB system involves lattice distortion resulting from the randomness of cationic arrangements. The relaxation behavior in SBN, particularly in $Sr_{0.67}Ba_{0.33}Nb_2O_6$ material, is attributed to the size mismatch between Sr and Ba ions at A2 sites, displacing adjacent oxygen atoms and thereby inducing a net polarization effect[6]. Additionally, the specific ion occupancy in the crystal structure reveals that the ferroelectric relaxation phase transition in SBN occurs when more than half of the $Sr^{2+}$ occupy the A2-site. The transition from normal ferroelectric to relaxor state is gradual, occurring in compositions with $x \geq 0.60$ in single crystals and $x \geq 0.53$ in ceramics[7].

To enhance the energy storage performance in dielectric materials, researchers utilized strategies such as refining grain morphology or grain orientation at a mesoscopic scale[8,9] as well as implementing domain engineering at a microscopic level[10,11]. Despite these efforts, meeting the stringent requirements of modern devices for high-performance and reliable energy storage capacitors remained a formidable challenge. Entropy modulation has emerged as a promising alternative strategy to augment energy storage capacity[12]. Configurational entropy provides a quantitative measure to assess the heterogeneity of local components, thereby facilitating the optimization of energy storage performance through the manipulation of relaxation characteristics in ferroelectrics[13]. The configurational entropy of the oxide system can be augmented by increasing the number of elements randomly distributed on the same lattice site[14]. The molar configurational entropy ($\Delta S_{config}$) of oxide systems can be calculated according to Eq. (1)[15]

$$S_{config} = -R\left[\left(\sum_{i=1}^N x_i \ln x_i\right)_{cation-sit} + \left(\sum_{j=1}^M x_j \ln x_j\right)_{anion-sit}\right] \quad (1)$$

where $x_i$ and $x_j$ represent the mole fraction of elements present in the cation and anion sites, respectively, and $R$ is the universal gas constant. As per empirical classification, materials can be categorized based on their configurational entropy. Those with $\Delta S_{config} \geq 1.5R$ can be classified as "high entropy", while materials with $1.5R > \Delta S_{config} \geq 1.0R$ fall into the "medium entropy" category, and materials with $\Delta S_{config} < 1.0R$ are classified as "low entropy" systems[16].

Dielectrics with high entropy possess the distinctive capacity to achieve disordered polarization configurations via meticulously designed local structures[12]. The high entropy effect augments system disorder by creating a solid solution of multi-component elements, effectively regulating the stability of the entropy-dominated phase. Meanwhile, the increase in atomic disorder induces notable lattice distortion, impedes element diffusion, and triggers a synergistic cocktail effect from multi-component properties[14]. Due to these characteristics of high-entropy materials, the high entropy strategy has been applied to a variety of material structure systems to enhance energy storage performance, including perovskite structure[17], bismuth layer structure[18], pyrochlore structure[19], and tungsten bronze structure[20]. For example, Guo et al.[21] obtained a $W_{rec}$ of 10.7 J cm$^{-3}$ and a efficiency of 89% in $Li_2CO_3$-doped $Bi_{0.2}Na_{0.2}Ba_{0.2}Sr_{0.2}Ca_{0.2}TiO_3$ high-entropy ceramics with a perovskite structure. The coexistence of randomly distributed A-site ions and B-site ions facilitates intricate interactions, resulting in the emergence of atomic-scale low crystallographic symmetries. This involved generating multiphase nanoclusters to achieve extremely small polar nanoregions, thereby enhancing the breakdown field, delaying polarization saturation, and significantly improving the energy storage performance[22].

Band gap engineering involves adjusting properties through doping to modify the material's band gap. Increasing the band gap (i.e., raising the energy required for electrons to move from the valence band to the conduction band) can heighten the dielectric breakdown strength by reducing the intrinsic carrier concentration and conductivity, thereby increasing the electric field required for intrinsic breakdown[23]. This strategy not only significantly improved the energy storage performance but also achieved excellent high-temperature stability.

In this study, we enhance the energy storage performance of tetragonal tungsten bronze structure ceramics, specifically $Ba_{0.4}Sr_{0.6-x}Ca_xNb_{2-x}Ta_xO_6$ (BSCNTx, x = 0,0.15,0.3,0.45) ceramics, by employing a combination of high-entropy strategy and band gap engineering. Incorporating oxides with high band gaps, such as CaO and $Ta_2O_5$, can elevate the overall band gap of the compound, thereby enhancing the breakdown field. Concurrently, the heightened configurational entropy (from 0.68R to 1.54R) induces the cocktail effect, which is harnessed to optimize the energy storage performance of the $Ba_{0.4}Sr_{0.6}Nb_2O_6$ ceramics. This optimization culminates in exceptionally high energy storage properties, characterized by a $W_{rec}$ of 8.9 J cm$^{-3}$ and an impressive efficiency of 93%. Crucially, the BSCNT0.30 ceramics can maintain a $W_{rec}$ of up to 4.9 J cm$^{-3}$ with a high $\eta$ of 89% even at elevated temperature of 180 °C. Coupled with its remarkable frequency insensitivity and fatigue resistance, this system shows significant potential for application in advanced pulsed power devices operating in harsh environments.

## Results and discussion

Figure 1a shows the *P-E* loops of different compositions at their respective maximum electric field, i.e., the breakdown field. With the increase of configurational entropy, the breakdown field obviously increases from 360 kV cm$^{-1}$ to 700 kV cm$^{-1}$ at high entropy of 1.51R, above which, the breakdown field decreases, this is due to the fact that the excessive addition of Ca/Ta introduces impurities that deteriorate the breakdown field. Meanwhile, the $P_r$ value is negatively correlated with configurational entropy, decreasing from 4.4 to 1.6 µC cm$^{-2}$. The relationship between the corresponding energy storage performance and configuration entropy is established, as illustrated in Fig. 1b. The transition from low entropy to high entropy is evident in the enhancement of energy storage performance, increasing from 4.3 J cm$^{-3}$ (BSCNT0) to 8.9 J cm$^{-3}$ (BSCNT0.30), representing an increase exceeding 100%. Meanwhile the efficiency also improves significantly, rising from 81% to 93%. Despite BSCNT0.45 exhibiting a

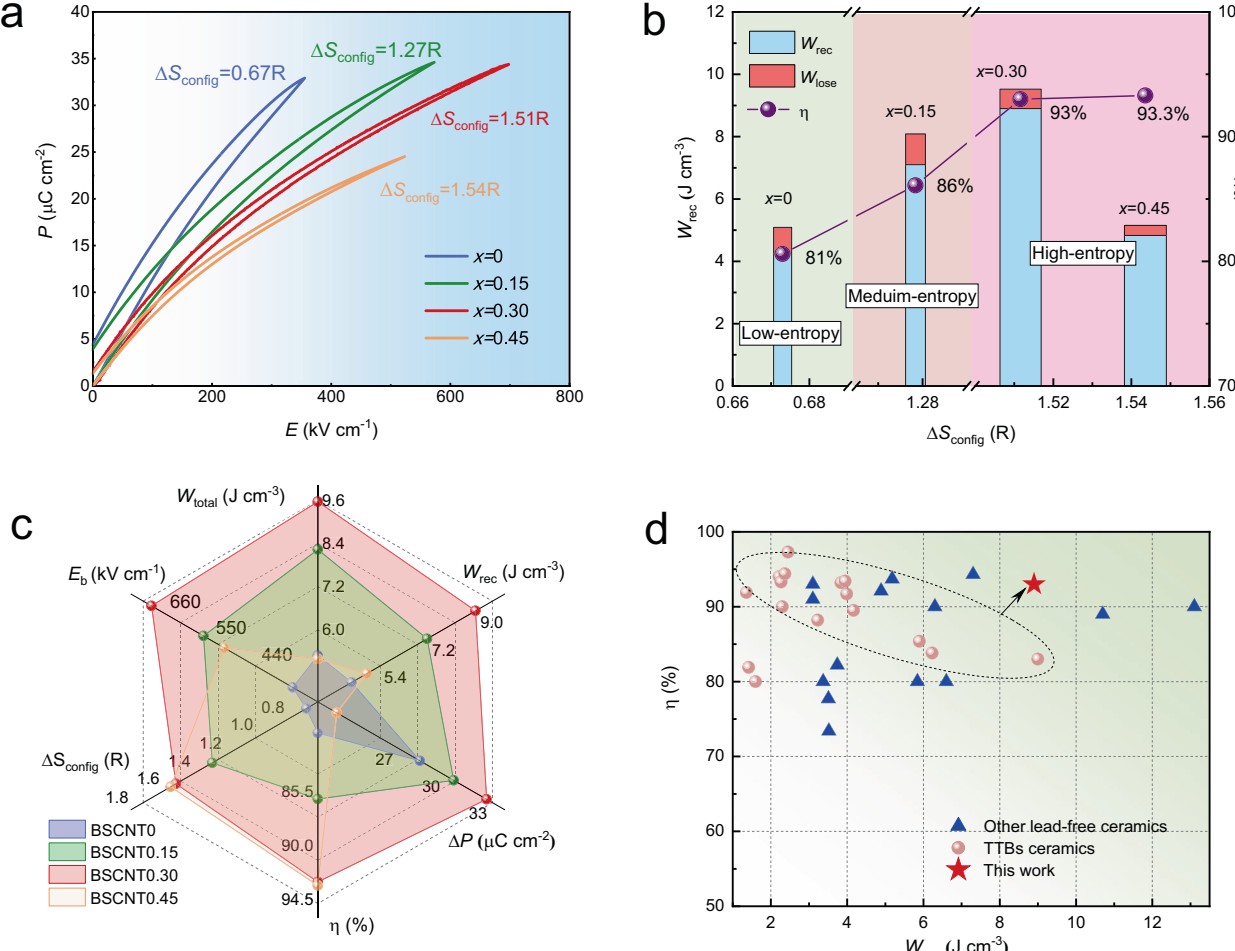

**Fig. 1 | Energy storage performance of BSCNT*x* ceramics. a** Unipolar *P-E* loops under maximum electric field of BSCNT*x* ceramics. **b** Energy density and efficiency, calculated based on the *P-E* loops of BSCNT*x* ceramics at the maximum electric field, presented as a function of conformational entropy. **c** Comparison of $E_b$, $\Delta S_{config}$, and energy storage performance of BSCNT*x* ceramics. **d** Comparison of the energy storage performance ($W_{rec}$ and $\eta$) of the BSCNT0.30 ceramics with other representative lead-free ceramics and TTBs ceramics reported previously.

high efficiency of 93.3%, its low breakdown field and polarization limit its energy storage potential.

To investigate the effect of configurational entropy on the electric field response of BSCNT*x* ceramics, the bipolar *P-E* loops of BSCNT*x* ceramics with different configurational entropies were measured under the same electric field (150 kV cm⁻¹), as shown in Fig. S1a. The corresponding *I-E* curves and polarization changes are also illustrated in Fig. S1b, c. With the increase in configuration entropy, both $P_{max}$ and $P_r$ exhibit a gradual decrease, indicating weakened ferroelectricity and enhanced relaxation characteristics, which can be substantiated by the gradual flattening of the switching current peak, thereby enabling higher $W_{rec}$ and $\eta$[24]. To visually demonstrate the impact of entropy on the energy storage capabilities of BSCNT*x* dielectric ceramics, Fig. 1c compiles data on breakdown strength, ΔP, energy storage performance, and configurational entropy of various compositions. It is evident that BSCNT0.30 ceramics exhibit the most favorable comprehensive energy storage performance, attributed to their exceptionally high $E_b$ and ΔP resulting from elevated configurational entropy. Additionally, Fig. 1d compares the energy storage performance of our study with that of other recently published lead-free dielectric ceramics and TTBs ceramics, from which our findings exhibit a leading comprehensive energy storage performance in TTBs system.

To delineate the influence of configurational entropy on the crystal structure of BSCNT*x* samples, Fig. 2a depicts the XRD pattern of the BSCNT*x* ceramics. It is evident that, with the increase in *x*, the structure of the tetragonal tungsten bronze is maintained; however, some impurities are observed in BSCNT0.45. The secondary phase is mainly composed of $CaNb_2O_6$ (PDF#71-2406) and/or $Ba_6Nb_2O_{11}$ (PDF#46-0938). These impurities produced by BSCNT0.45 is due to the small ionic radius of $Ca^{2+}$, which tends to enter the A1 position of the quadrilateral and difficult to enter the A2 position, resulting in limited solubility[25]. In particular, one third of the A-site is occupied by A1-site, while two-thirds are filled by A2-site. Additionally, in the unfilled structure, one-sixth of the A-sites remain vacant and are randomly distributed between A1 and A2 sites. Therefore, even if all vacancies and $Sr^{2+}$ ions are located exclusively in the A2-site, the content of $Ca^{2+}$ can only account for a maximum of one-third of the A-site, corresponding to *x* = 0.4. Beyond this range will result in the inability to maintain a single phase[25,26]. In Fig. 2b, a locally amplified XRD pattern is presented. As configurational entropy increases, the peaks corresponding to (211), (530), and (620) gradually shift to the right, indicating lattice shrinkage. Figure 2c illustrates the projection of BSCNT0 and BSCNT0.30 along crystallographic [001] direction. The lattice shrinkage is due to the smaller radius of $Ca^{2+}$(1.34 Å) compared to $Sr^{2+}$ (1.44 Å) and $Ba^{2+}$ (1.61 Å). Moreover, despite the same ionic

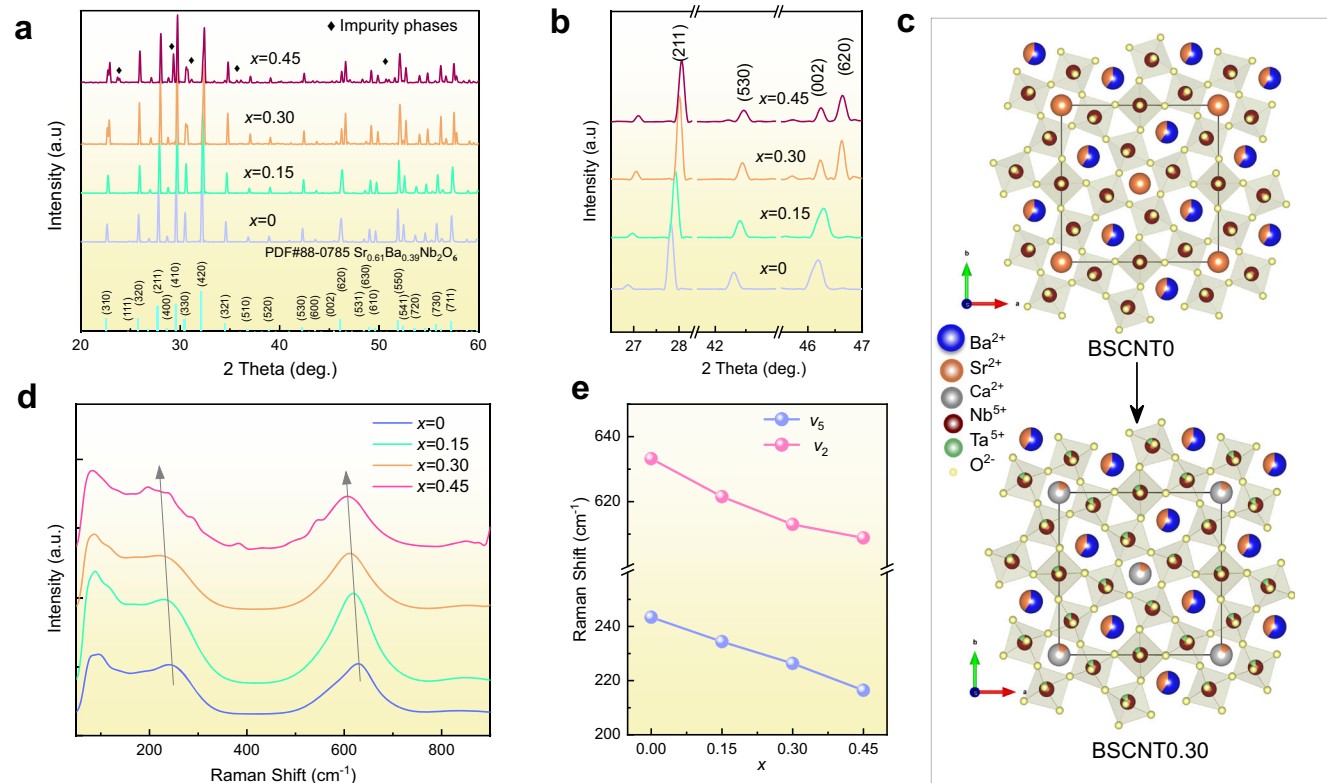

**Fig. 2 | Crystal structure analysis of BSCNT*x* ceramics. a** X-ray diffraction patterns of the ceramics with increasing *x* and (**b**) enlarge view of selected diffraction peak. **c** Projection of BSCNT0 and BSCNT0.30 unit taken along [001] (*P4bm*, solid line). **d** Raman spectroscopy of BSCNT*x* ceramics. **e** The variation in wavenumber $v_2$ and $v_5$ Raman vibration modes were divided by Gaussian-Lorentzian function.

radius of $Ta^{5+}$(0.64 Å) and $Nb^{5+}$ (0.64 Å), the increased electronegativity difference between $Ta^{5+}$ and $O^{2+}$ compared to $Nb^{5+}$ leads to an increase in the polarity of the B-O bond. This, in turn, shortens the B-O covalent bond length, resulting in a contraction of the cell volume[27]. It is evident that as the configurational entropy increases, a variety of elements occupy both the A-site and B-site, resulting in increased system disorder. The Rietveld refinement results of XRD data for BSCNT0 and BSCNT0.30 are illustrated in Fig. S2. The mean ionic radii of the distinct A-sites within the respective components were computed based on the cationic position and proportion (Tables S1 and S2). The results showcase that the mean ionic radius of BSCNT0 is 1.44 Å and 1.54 Å at A1-sites and A2-sites respectively. For BSCNT0.30, on the other hand, the mean ionic radius is 1.36 Å at the A1-sites and 1.55 Å at the A2-sites. The augmented disparity in radius between A1 and A2 sites is expected to induce lattice distortion, which will affect the polarization and corresponding energy storage performance.

To understand the changes in the local structure of BSCNT*x* ceramics with various configuration entropy, the Raman spectrum was measured, as shown in Fig. 2d. Consistent with previous reports, three main regions are observed in the wave number range of 20 to 900 cm$^{-1}$, (ref. 28). The low wave number region corresponds to the vibration of A-site cations (less than 200 cm$^{-1}$), the medium wave number region is related to O-B-O bending vibrations (200–400 cm$^{-1}$), and the high wave number region above 400 cm$^{-1}$ is closely related to the B-O stretching vibrations[29]. The sample exhibits a broad and diffuse Raman spectrum mode, which represents the enhancement of elemental disorder and the reduction of unit cell polarity caused by the random distribution of cations at the A and B sites[30]. Similar to the results of XRD pattern, the Raman spectrum of BSCNT0.45 shows multiple anomalous peaks, which are caused by the secondary phases.

At the same time, for a more intuitively understand of the local structure evolution, Gaussian peak fitting was performed on the BSCNT*x* Raman diagram, as illustrated in Fig. S3. The changes in the wave numbers of the $v_2$ and $v_5$ peaks with *x* are shown in Fig. 2e. The positions of bands $v_2$ and $v_5$ move toward lower wave numbers with increasing configurational entropy, indicating a decrease in unit cell polarity[31].

The breakdown strength stands as a critical factor influencing the energy storage performance of dielectric ceramics. In order to accurately evaluate the breakdown strength of BSCNT*x* ceramics, breakdown experiments were carried out on 10 samples of each component, and the standard Weibull distribution analysis was calculated according to the results. The fitting results and reliable breakdown electric fields of different components are illustrated in Fig. 3a. The parameter β, denoting the Weibull modulus, serves as an index reflecting the dispersion within the distribution of data.

It is reported that the breakdown strength is closely related to the grain size[32], bandgap[21], space charge[33]. Figure S4a–d present scanning electron microscopy (SEM) images showcasing the surface morphology, along with an illustration demonstrating the statistical distribution of grain sizes ($G_a$) in the corresponding samples. The average grain size of BSCNT*x* decreases with increasing entropy. Specifically, for BSCNT0.45 ceramics, the average grain size is reduced by 50% compared to BSCNT0, decreasing from 3.4 μm to 1.7 μm. Given their remarkable refractory behavior, oxides of Ta and Ca are extensively employed as grain growth inhibitors to mitigate the average grain size[24,34]. Furthermore, the lattice distortion induced by the elevated configurational entropy can substantially impede atomic diffusion, thereby further diminishing the average grain size thus higher grain boundary density[35]. Notably, grain boundaries with higher resistivity play an important role in preventing electric breakdown by absorbing

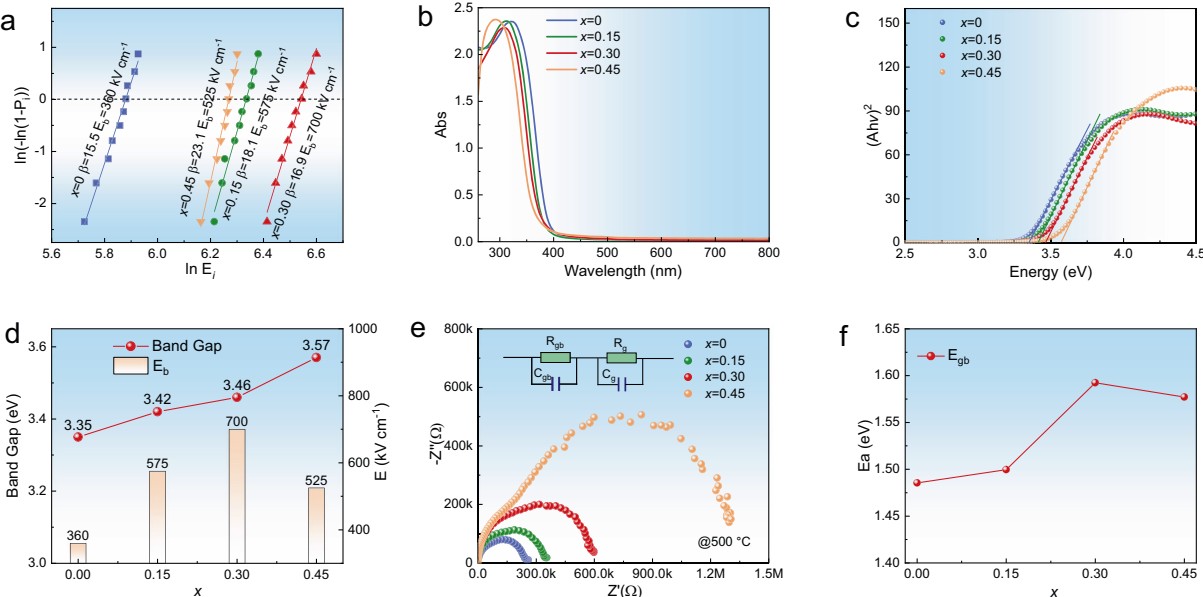

**Fig. 3 | Breakdown strength, band gap and impedance spectrum analysis of BSCNTx ceramics. a** Weibull distribution analysis of the breakdown strengths of the ceramics. The $\beta$ is the Weibull modulus. The sample thickness is 70–85 μm and the electrode area is 0.7 mm². **b** UV-vis absorption spectra of BSCNTx ceramics. **c** The Tauc plots of all BSCNTx samples. **d** Band gap and breakdown strength corresponding to different compositions. **e** Impedance spectroscopy of BSCNTx at 500 °C (the inset showing the equivalent circuit proposed for impedance data fitting). **f** The fitting grain boundary activation energy values ($E_{gb}$) versus the x.

the energy from electric tree expansion within the grains[36]. Smaller grain sizes correspond to a higher grain boundary density, contributing to an obvious negative correlation between the average grain size and breakdown field, that is $E_b \propto (G_a)^{-1}$,[32]. Meanwhile, Fig. S4e clearly illustrates that the BSCNT0.30 sample exhibits a uniform distribution of elements without any noticeable segregation, confirming the successful incorporation of $Ca^{2+}$ and $Ta^{5+}$ into the lattice. This homogeneous composition, coupled with the refined grain size, contributes to enhancing breakdown strength by altering the electric field distribution within ceramic under applied electric field[37].

The intrinsic breakdown mechanism of dielectrics primarily involves electron breakdown, where the band gap serving as the decisive factor. Figure 3b illustrates the UV-visible absorption spectrum of BSCNTx ceramics. while Fig. 3c presents the determination of the direct bandgap of BSCNTx ceramics using the Tauc plot method. The introduction of $Ca^{2+}/Ta^{5+}$ can elevate the band gap by altering the minimum value of the conduction band[38]. As anticipated, owing to the wider band gap of $Ta_2O_5$ (4.3 eV)[39] and CaO (7.1 eV)[40] in comparison to $Nb_2O_5$ (3.4 eV)[41] and SrO (6.1 eV)[42], the band gap of BSCNTx eventually increases from 3.35 eV (BSCNT0) to 3.46 eV (BSCNT0.30), as shown in Fig. 3d, leading to a significant enhancement in the breakdown strength.

To delve deeper into the high $E_b$ exhibited by the BSCNTx ceramics, complex impedance spectra are presented in Fig. S5a–d, covering temperatures ranging from 450 °C to 650 °C. Figure 3e gives the impedance spectrum of BSCNTx at 500 °C. It is apparent that as the configurational entropy increases, the radius of impedance spectrum arc gradually increases, highlighting a gradual rise in resistance. This observation implies that higher configurational entropy enhances the insulation performance of ceramics. The block ceramic-electrode system can be simplified as a brick structure consisting of grain boundaries and grains. Two sets of equivalent R-C circuit elements, representing grains and grain boundaries, were connected in series to fit the impedance data, as illustrated in the inset of Fig. 3e. The activation energy of conduction for the grain boundary ($E_{gb}$) of the BSCNTx ceramic system can be calculated by the Arrhenius formula[43],

as showcased in Fig. 3f. The increase in $E_{gb}$ indicates a decrease in the concentration of oxygen vacancies at these grain boundaries, resulting in higher resistivity of the boundaries. The activation energy of conduction was positively correlated with the breakdown strength[44]. By incorporation of $Ca^{2+}$ and $Ta^{5+}$ into $Ba_{0.4}Sr_{0.6}Nb_2O_6$ ceramics, $E_{gb}$ exceeding 1.58 eV can be achieved in BSCNT0.30 samples, significantly enhancing the breakdown strength. Nevertheless, in the BSCNT0.45 ceramics, the solid solubility limitation of $Ca^{2+}$ ions within the SBN matrix gives rise to secondary phases, such as $Ca_2Nb_2O_6$ and/or $Ba_6Nb_2O_{11}$. Notably, oxygen vacancy exists in $Ba_6Nb_2O_{11}$ to maintain electrical neutrality[45]. Since the activation energy inversely correlates with the concentration of oxygen vacancies[44], BSCNT0.45 exhibits a lower activation energy, which is further supported by the decreased breakdown field observed in BSCNT0.45.

In addition to breakdown strength, the high saturation polarization and low remanent polarization of relaxor ferroelectrics are the favorable characteristics for their applications in energy storage. Therefore, enhancing the relaxation characteristics of dielectrics is a crucial approach to optimize their energy storage performance, which can be achieved by high entropy strategy[2]. To investigate the evolution of relaxation behavior in the BSCNTx ceramics, the relative permittivity ($\varepsilon_r$) and loss tangent ($\tan\delta$) were measured over a temperature range of −100 to 250 °C and a frequency range from 10 Hz to 2 MHz, as depicted in Fig. S6a–h. The observed less steep dielectric peaks, along with significant frequency dispersion behavior, suggest a typical dispersion phase transition characteristic of relaxor ferroelectrics[46]. In Fig. S7a, the temperature dependence of the $\varepsilon_r$ and $\tan\delta$ of BSCNTx samples at 1 kHz is presented. As configurational entropy increases, the maximum dielectric constant in BSCNTx samples decreases, and the temperature corresponding to the maximum dielectric constant ($T_m$) shifts to a lower temperature. Frequency dispersion behavior, indicating an increase in $T_m$ with rising frequency, is a crucial characteristic of relaxor ferroelectrics. As illustrated in Fig. S7b, $\triangle T_m$ ($T_{m\ 2MHz}$-$T_{m\ 10Hz}$) notably rises from 17 K to 59 K with an increase in configurational entropy, signifying that increased entropy is beneficial for enhancing relaxor component in BSCNTx. Furthermore, the semi-

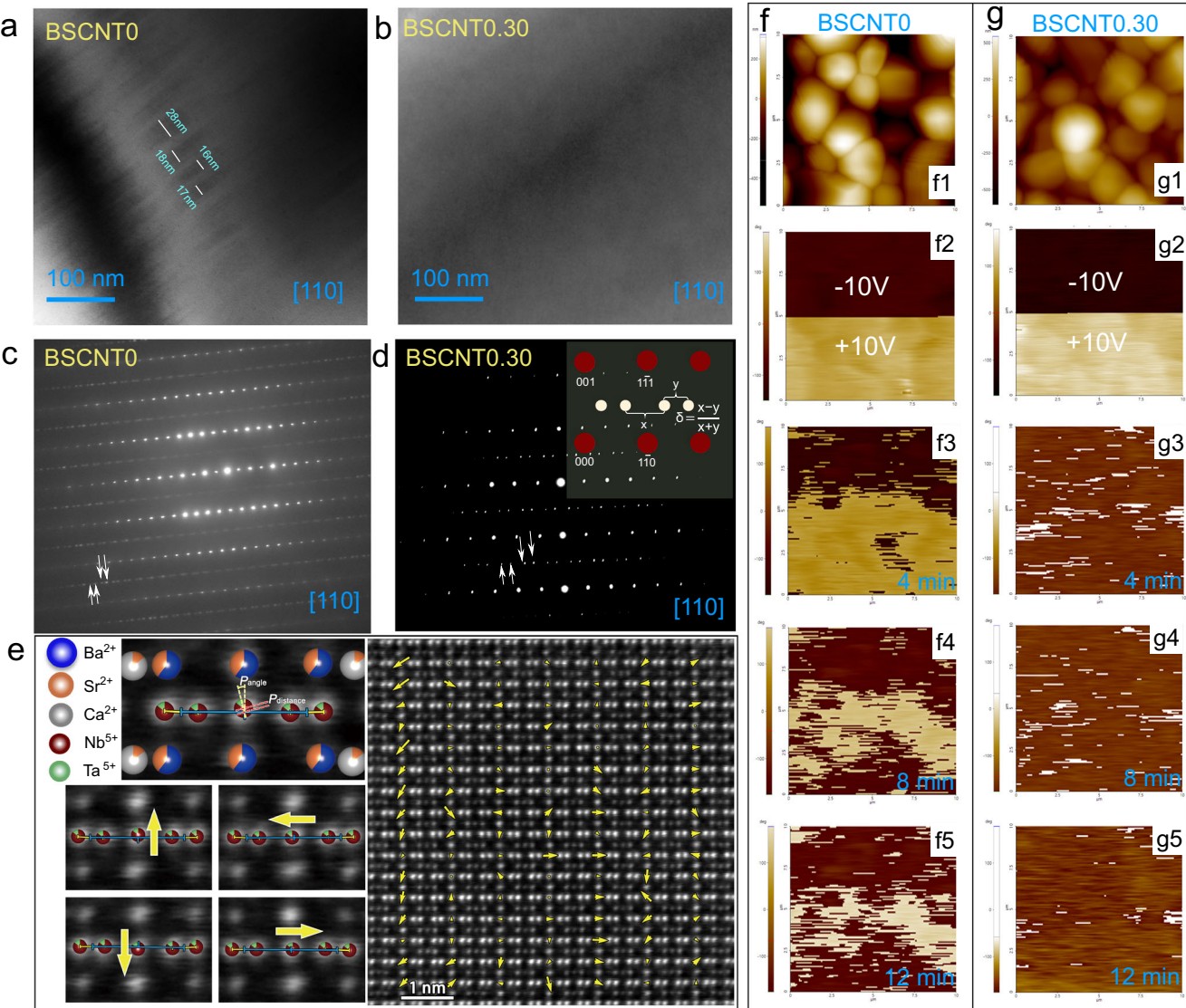

**Fig. 4 | TEM characterization and out-of-plane PFM phase images of BSCNT0 and BSCNT0.30 ceramics. a**, **b** TEM image patterns take along [110] zone from BSCNT0 and BSCNT0.30 samples, (**c**, **d**) SAED patterns take along [110] zone from BSCNT0 and BSCNT0.3 samples. The inset of (**d**) displays the measurement schematic for determining the incommensurate superlattice incommensurability parameter δ. The white arrows indicate the appearance of the satellite reflections for incommensurate modulation in **b** and **d**. **e** Atomic-resolution HAADF-STEM image along [110] of the BSCNT0.30 ceramic, and the arrows show the displacement of B-site cations corresponding to different polarization directions. **f**, **g** Out-of-plane PFM phase images of samples BSCNT0 and BSCNT0.30 after poling treatment with ±10 V and then measuring after a certain relaxation time.

quantitative evaluation of the relaxation properties can be achieved by fitting the diffusion coefficient γ based on the modified Curie-Weiss equation[47], exhibiting an increasing trend from 1.35 to 1.73, as given in Fig. S7c, indicating a stronger relaxation behavior in high entropy BSCNT0.30 ceramics. The introduction of multiple elements into equivalent lattice positions via the high entropy strategy increases chemical complexity, disrupts long-range ordering in the ferroelectrics, and enhances the relaxor characteristics of BSCNT0.30[48].

The relaxation dynamic behaviors in BSCNT$x$ ceramics were assessed using the Vogel-Fulcher model[49]. The fitting outcomes are demonstrated in Fig. S6, while the computed activation energy ($E_a$) is depicted in Fig. S7d. The $E_a$ escalates from 0.013 eV (BSCNT0) to 0.073 eV (BSCNT0.30) with increasing configurational entropy. The increase in activation energy suggests a weakening of the coupling between polar clusters in the system, making it more challenging to form ordered polarized regions[50]. This phenomenon presents an advantage for energy storage, as it corresponds to delayed

polarization saturation under high electric fields and maintain a thinner $P$-$E$ loop[49].

To confirm the weakly coupled features of polar clusters in the BSCNT$x$ ceramics and elucidate the origin of relaxation, field emission transmission electron microscopy (TEM) observations were conducted on BSCNT0 and BSCNT0.30 samples, as depicted in Fig. 4a, b. The BSCNT0 sample (Fig. 4a) exhibits noticeable long-range continuous polarization, manifesting as ferroelectric domains with widths extending up to tens of nanometers. Dramatically divergent, the BSCNT0.30 sample lacks such discernible ferroelectric domain structures, as shown in Fig. 4b. Figure 4c, d depict the corresponding selected area electron diffraction (SAED) patterns for BSCNT0 and BSCNT0.30 along the [110] zone axis. Alongside the Bragg reflections corresponding to the TTB structure, additional superlattice spots associated with structural modulation (indicated by white arrows) emerge in the SAED patterns, signifying the presence of incommensurate modulation in BSCNT$x$ ceramics at room temperature[51]. The

presence of a commensurate modulation is strongly associated with the long-range ordering and demonstrates ferroelectric properties. Conversely, the existence of an incommensurate modulation suggests a pronounced relaxation characteristics[52]. The incommensurate modulation wave vectors were calculated as $(1/4 + \delta)(\textbf{a-b})+1/2\textbf{c}$, where **a, b**, and **c** are the vectors in reciprocal space and $\delta$ is the parameter that describes the deviation from commensurate periodicity[53]. $\delta$ is calculated using the formula $\delta = (x - y)/(x + y)$, where $x$ and $y$ represent the distances from the adjacent incommensurate diffraction spots, as determined by measuring the positions of weak reflections[52]. The measurement method for determining incommensurability parameter $\delta$ is presented in inset of Fig. 4d. It can be obtained by calculating the weak reflections: $\delta_{BSCNT0} = 0.18 \pm 0.04$ (300 K), $\delta_{BSCNT0.30} = 0.28 \pm 0.06$ (300 K). The increased incommensurability parameter arises from the A-site disorder, consequently leading to diffused satellite reflection points[53]. Hence, the larger $\delta$ observed in BSCNT0.30 is ascribed to the random occupancy of $Ba^{2+}$, $Sr^{2+}$, and $Ca^{2+}$ in the A-site, accounting for the enhanced relaxation behavior[54]. Simultaneously, the addition of $Ta^{5+}$ causes unavoidable distortion of the oxygen octahedron, leading to local lattice distortion. This suggests that B-site ion doping tends to promote incommensurate modulation[31].

To further illustrate the presence of weakly coupled polar nanoregions (PNRs) and the influence of high entropy effect on local polarization distribution, high-angle annular dark-field scanning transmission electron microscopy (HAADF-STEM) experiments were conducted on the high-entropy BSCNT0.30 sample, as depicted in Fig. 4e. In the HAADF-STEM images, A-site atoms were observed to occupy highly symmetric positions, showing minimal relative movement among them. In contrast, B-site atoms exhibited multi-directional displacement from the central position between two neighboring atoms, as evidenced by the lower left images in Fig. 4e. Since the polarization of TTBs stems from the displacement of B-site ions[55], examining the movement of the central B-site cation in relation to adjacent B-site cations allows for the determination of local polarization direction. As illustrated in Fig. 4e, the displacement of B-site

atoms in the BSCNT0.30 sample appeared highly disordered, lacking consistent displacement directions. This disorder results in the formation of highly segmented PNRs and weak coupling effects among them. Such a disordered structure significantly impedes the long-range order of the ferroelectrics, ultimately contributing to the enhanced relaxation and thereby excellent energy storage performance[18]. Meanwhile, such localized ordered structures are more conducive to polarization realignment under the influence of an applied electric field, reduce the domain-switching energy barriers, resulting in reduced heat generation and thereby enhancing thermal breakdown strength[36].

The dynamic response of PNRs to external electric fields is closely related to the energy storage properties of dielectric materials[56], which was investigated using piezoelectric response force microscopy (PFM). As shown in Fig. 4f1-f5 and g1-g5, after being written with DC voltages of +10 V and −10 V, respectively, read scans were performed every 4 min. It was observed that the polarization of BSCNT0.30 returned to its initial highly random state much faster compared to BSCNT0. Notably, as depicted in Fig. 4g5, BSCNT0.30 exhibited almost no identifiable PNRs after 12 min of polarized writing, highlighting the weak coupling between polar clusters. This suggests the presence of more dynamic PNRs in BSCNT0.30 ceramics, a significant characteristic of enhanced relaxation. The rapid reversibility of PNRs leads to reduced $P_r$, delayed polarization saturation, and ultimately exceptional energy storage performance[57].

Given the advantage of excellent high-temperature resistance in dielectric ceramics, the temperature stability of their energy storage performance is of paramount importance. To assess the energy storage performance of BSCNT0.30 at different temperatures, the $P$-$E$ loop was tested at 500 kV cm$^{-1}$ across temperature range of 30 °C to 180 °C, as shown in Fig. 5a. The corresponding energy storage performance at different temperatures is depicted in Fig. 5b. Across the tested temperature range, $W_{rec}$ and $\eta$ consistently maintained high values, with $W_{rec}$ ranging from 5.1 to 4.9 J cm$^{-3}$ and $\eta$ ranging between 89-92%. The change rates were less than 5% and 3%, respectively. This

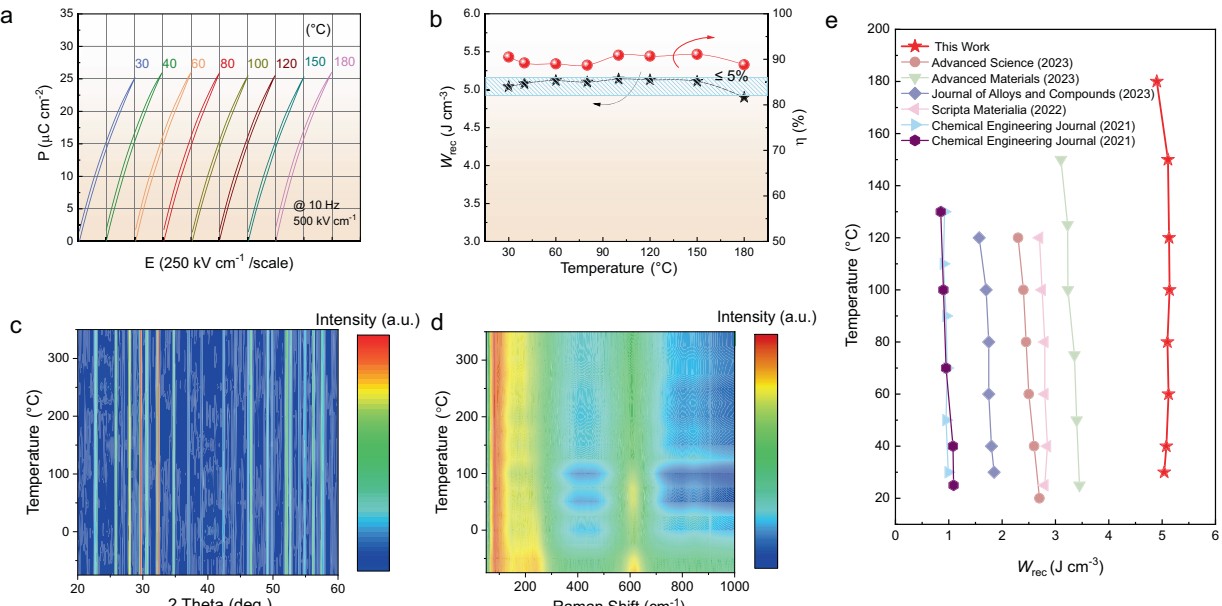

**Fig. 5 | Temperature stability of energy storage performance of BSCNT0.30 ceramics. a** $P$-$E$ loops with different temperatures at 500 kV cm$^{-1}$. **b** Variation of recoverable energy density and efficiency with temperature. **c** XRD patterns of BSCNT0.30 ceramics at different temperature. **d** Raman spectra of BSCNT0.30 ceramics at different temperature. **e** Comparison of this work with other recently published TTBs ceramics for variable temperature energy storage performance.

outcome illustrates that, owing to the high-entropy effect, the energy storage performance of BSCNT0.30 exhibits excellent temperature stability. To delve deeper into the reason behind the high-temperature stability of BSCNT0.30, its structural changes with temperature were tested. Figure 5c displays the XRD pattern from −70 to 350 °C, revealing minimal changes in the peak position and width of each diffraction peak. Similarly, as shown in Fig. 5d, the Raman test pattern obtained in the temperature range of −70 to 350 °C demonstrates that although the peak intensity decreases at the wave number of ~600 cm$^{-1}$, the peak position remains largely unchanged. The excellent temperature stability of BSCNT0.30's energy storage performance is attributed to the temperature insensitivity of its crystal structure (as shown in Fig. 5c, d) and the ultra-low dielectric loss of <0.003 across the temperature range up to 180 °C. Moreover, the complex impedance spectrum indicates that BSCNT0.30 maintains extremely high resistance (~2MΩ) even at 450 °C, demonstrating its excellent high-temperature insulation properties. The excellent temperature stability may also be attributed to the superparaelectric (SPE) state of BSCNT0.30. The temperature-driven SPE in relaxors was report to maintain nonlinear polarization with high $P_{max}$ and minimal hysteresis, which is desirable for achieving both high $W_{rec}$ and $\eta$ in dielectric capacitors[58]. At 180 °C, the increase in $P_{max}$ is attributed to additional charge from increased leakage at high temperatures, while the increase in $P_r$ may result from conduction losses due to thermal stimulation, ultimately leading to lower efficiency[59].

Figure 5e illustrates the comparison of the recoverable energy storage density of the studied compositions with recently published TTBs ceramics across different temperature ranges[24,28,29,55,60,61]. In comparison to these works, the BSCNT0.30 exhibits higher energy storage density over a wider temperature range up to 180 °C, making it suitable for applications in high-temperature environments.

In addition to temperature usage range and stability of the energy storage performance, cycling reliability is another crucial consideration for real applications. Figure S8a, b depict the P-E loops of BSCNT0.30 after various cycles at 350 kV cm$^{-1}$. These loops maintain a slimmed shape with minimal hysteresis and high $P_{max}$ over the cycle range of $10^0$–$10^5$, accompanied by negligible changes in $W_{rec}$ (3-3.1 J cm$^{-3}$) and $\eta$ (93-96%), showcasing excellent fatigue resistance and superior cycling reliability. Figure S8c, d shows the P-E loop of BSCNT0.30 across various frequencies (5–500 Hz) and the associated energy storage performance at 350 kV cm$^{-1}$. The $W_{rec}$ and $\eta$ values remain stable at approximately 3 J cm$^{-3}$ and 87–93%, respectively, indicating excellent frequency insensitivity properties.

In practical applications, assessing the charge and discharge capabilities holds paramount importance. To evaluate the real charge and discharge performance of BSCN0.30 ceramics, both underdamped and over-damped charge and discharge tests were conducted. Additionally, the alterations in charge and discharge performance at various temperatures were thoroughly evaluated. The results are presented in Figs. S9 and S10. The findings reveal that BSCNT attains an impressive current density ($C_D$) of up to 1500 A cm$^{-2}$, a power density ($P_D$) of 280 MW cm$^{-3}$, a discharge energy storage density ($W_{diss}$) of 2.6 J cm$^{-3}$, and a discharge speed of $t_{0.9}$ = 69.4 ns. Notably, BSCNT0.30 showcases remarkable temperature stability, underscoring its immense potential for use in demanding operational environments, such as high-temperature settings.

In summary, by controlling configurational entropy and band gap within tetragonal tungsten bronze-structured BSCNT ceramics, we have successfully achieved a remarkable recoverable energy density ($W_{rec}$) of 8.9 J cm$^{-3}$ and an efficiency of 93% in the $Ba_{0.4}Sr_{0.3}Ca_{0.3}Nb_{1.7}Ta_{0.3}O_6$ TTBs ceramic. Notably, the excellent temperature stability enables BSCNT0.30 ceramics to maintain an energy storage density of greater than 4.9 J cm$^{-3}$ at 180 °C while achieving an efficiency of up to 89%. The comprehensive structural characterization

proves that the random cation occupancy caused by the high entropy effect breaks the long-range order of the ferroelectric, resulting in highly dynamic and weakly coupled polar nano regions. Meanwhile the high entropy effect restricts the grain growth and increases the overall resistivity of the ceramics, together with the increased band gaps, contributing to an extremely high breakdown strength. The optimized polarization ΔP behavior and increased $E_b$ are responsible for the greatly improved energy storage performance in the TTBs ceramics, hold great potential for energy storage application across a broad temperature range.

## Methods

### Ceramics fabrication
In this research, a series of BSCNT ceramics were fabricated through a high-temperature solid-state reaction method. The ceramics with the composition $Ba_{0.4}Sr_{0.6-x}Ca_xNb_{2-x}Ta_xO_6$ (BSCNTx, 0 ≤x ≤ 0.45) were developed using high-purity powders: $Ba_2CO_3$ (99.8%), $SrCO_3$ (99%), $CaCO_3$ (99.99%), $Nb_2O_5$ (99.9%), and $Ta_2O_5$ (99.9%). The powders were accurately measured based on their stoichiometric proportions and then combined with zirconia grinding media and anhydrous ethanol. This mixture was subjected to ball milling at 400 rpm for 12 h in a polyethylene vessel to achieve uniform mixing. Following this, the dried powder was pre-sintered at 1100 °C for 2 h and underwent a second ball milling under the same conditions. The resulting fine powder was then compacted into 10 mm diameter discs using cold isostatic pressing at 200 MPa for 60 seconds. The compacted samples were finally sintered in an alumina crucible at temperatures ranging from 1300 °C to 1370 °C for 2 h.

### Structural characterization
The crystal structures of samples were analyzed using X-ray diffraction (XRD, SMARTLAB, Japan) with Cu Kα-0.1541 nm. Surface morphology and elemental distribution were examined by a field emission scanning electron microscope (Gemini SEM 500, Carle Zeiss, Germany). Raman spectrometry (HR800, Horiba JOBIN YVON) with a wavelength of 532.3 nm was employed to acquire detailed information about doping effect on the structural change of BSCNT ceramics. The HAADF-STEM images were taken by an aberration-corrected (scanning) transmission electron microscope (JEM-ARM300F2, JEOL, Japan). UV-Vis absorption spectra were obtained by a UV-Vis Spectrometer (PerkinElmer Lambda 950).

### Electrical properties characterizations
The dielectric properties of the samples were measured using an LCR meter (E4980A, Agilent). The samples were polished to ~0.6 mm thick, coated with conductive silver paste and sintered at 550 °C for 30 min. The P-E hysteresis loops and current-electric field (I-E) curves under different electric fields were measured by Precision Premier II from Radiant Technologies, connected to a high voltage amplifier. The samples are polished down to 70–85 μm in thickness, then coated with platinum electrode with a thickness of ~40 nm and a diameter is 0.95 mm by magnetron sputtering for P-E hysteresis loops measurements. Diamond polishing liquids with particle sizes of 3 μ, 2 μ, and 0.5 μ were used for polishing to ensure low surface roughness. Piezo-response force microscopy (PFM) measurements were conducted using Park Systems XE7 AFM. Under-damped and over-damped (with a load resistance of 300 Ω) charge/discharge measurements were carried out by capacitor charge-discharge system (CFD-001, Gogo Instruments, China). The sample thickness for charging/discharging test is ~150 μm, with an electrode diameter of 2 mm.

## Data availability
The data that support the findings of this work are available within the article and its Supplementary Information file. Source data are provided with this paper.

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

## Acknowledgements

This work was supported by the National Natural Science Foundation of China (NSFC No. 52172125, 52302156), the National Natural Science Youth Foundation of China (Grant No. 12204393), the Major Science and Technology Project of Ordos City (2021EEDSCXQDFZ014), the Ordos Science and Technology Program (2022YY043), the Project from Xi'an Innovation Design and Research Institute Co., Ltd (20230134), The Shaanxi Province Qin Chuangyuan "Scientists plus Engineers" Team Project (2023KXJ-299), and the Research Grant Council of Hong Kong Special Administrative Region China (Project No. PolyU25300022). We thank the Instrument Analysis Center of Xi'an Jiaotong University for SEM and TEM measurements.

## Author contributions

This work was conceived and designed by Y.F.G., X.J.L., and S.J.Z. The samples were prepared by Y.F.G. The energy storage performance, dielectric properties, and temperature stability were tested, and the data were processed by Y.F.G. The surface polishing of the ceramics and the preparation of the electrodes were done by H.C.H. The complex impedance spectrum was tested, and the data were processed by J.W.M. The Raman spectrum and UV-visible absorption spectrum were measured and processed by Y.F.G. The SEM images were taken and processed by R.R.K. and X.P.Z. The TEM and HAADF-STEM images were taken and processed by Z.Z.S. and Z.B.C. The PFM images and XRD data were processed and analyzed by Y.F.G. and B.Y. The manuscript was drafted by Y.F.G. and Z.Z.S., revised by X.J.L., Z.B.C., and S.J.Z., and discussed with F.L. and J.Y.S. All authors contributed to the data analysis and discussion.

## Competing interests

The authors declare no competing interests.
