## [Peer Review File · Nature Communications]

Optimizing High-Temperature Energy Storage in Tungsten Bronze structured Ceramics via High-Entropy Strategy and Bandgap EngineeringREVIEWER COMMENTS

Reviewer #1 (Remarks to the Author):

The manuscript describes strategies to improve energy storage performance in BSCNT ceramics based on high entropy and band gap engineering, and demonstrates its powerful advantages to readers. The performance parameters are attractive enough, whether it is perovskite or tungsten bronze. The experimental data are abundant and sufficient to support most of the authors' conclusions, however, there are still confusing aspects of the scientific explanation. I, therefore, recommend that the manuscript be significantly revised by the authors and reviewed again for consideration for publication by Nature Communications.

* The important description of previous results is omitted in the introduction. The structural and electrical behavior of $Sr_xBa_{1-x}Nb_2O_6$ bronze has been studied for many years, with important contributions from institutions including the National Institute of Standards and Technology and Penn State University. In this regard, the authors should enrich the overview of SBN structural complexity (as stated by the authors in Page 4, lines 62-63) and basic dielectric properties. Please refer to 10.1063/1.1657277 and 10.1021/acs.chemmater.9b04122, etc.

* Page 11, lines 190-191. The authors claim that "...the Sr^{2+} that can occupy both A1 and A2, the disparity between A1 and A2 positions increases, leading to further lattice inhomogeneity." The specific meaning of "lattice inhomogeneity" here should be pointed out. In addition, why do differences in lattice site size affect "lattice inhomogeneity"?

* Page 13, lines 229-231. The statement "This suppression of grain growth is attributed to the sluggish diffusion effect observed in high-entropy materials" needs to be corrected. In fact, Ca or Ta are very common as grain growth inhibitors. Experimental phenomena, therefore, cannot be attributed solely to changes in entropy.

* Page 14, lines 250-252, and Page 15, lines 269-271. Without giving the specific properties of the impurity phase, it is believed that its damage to performance is flawed.

* Page 18, lines 323-325. The presence of incommensurability may not necessarily be related to relaxation behavior, as is observed in antiferroelectrics (10.1103/PhysRevB.72.024102), but simply exists as a structural feature that may be related to ion displacements. In addition, the authors ignored the impact of B site disorder (as mentioned by the authors in the previous section) on δ .

* The results of HAADF-STEM appear to provide a deeper understanding of the atomic structure of $Sr_xBa_{1-x}Nb_2O_6$.

* The most important issue is that the manuscript appears to only demonstrate the impact of bandgap engineering on energy storage performance. I carefully checked the authors and literature in similar fields, and found that the change in the dielectric behavior of BSCNT is very similar to the doping of a single element (10.1002/adma.202310559). The authors should further emphasize the important role of high entropy. Furthermore, classifying BSCNT as the high-entropy ceramics seems a bit reluctant, especially without giving detailed structural parameters.

Reviewer #2 (Remarks to the Author):

This paper shows that $Ba_{0.4}Sr_{0.3}Ca_{0.3}Nb_{1.7}Ta_{0.3}O_3$ with a tetragonal tungsten bronze-type structure has a high energy density and is an excellent dielectric material for energy applications.

The superiority of this material over other lead-free materials in terms of energy density,

energy storage efficiency, and the temperature dependence should be recognized as engineering, but the chemical composition system of the material is not new. I think that novel concepts and design guidelines that are generally applicable to tungsten bronze-type compounds are needed to be published in Nature commun. The authors explain various phenomena as effects of high entropy, assuming that high entropy is important for obtaining high energy density, but some of the explanations are not satisfactory.

(1) In this study, Sr and Nb are partially replaced by Ca and Ta in order to increase the entropy. The relaxor-like change can certainly be explained by the entropy increase, but there is no evidence that the dielectric breakdown strength is an effect of high entropy. In the case of perovskite-type oxides, the dielectric breakdown strength is more likely to be an elemental effect derived from Ca and Ta, as CaTiO₃ and KTaO₃ have higher dielectric strength than SrTiO₃ and KNbO₃, respectively. There is also no evidence that the smaller grain size is also due to a high entropy effect; the introduction of Ca and Ta may simply increase the melting point.

(2) The maximum polarization value of Ba_{0.4}Sr_{0.3}Ca_{0.3}Nb_{1.7}Ta_{0.3}O₃ is not very large for a tetragonal tungsten bronze-type oxide, and the high energy density is mostly due to the high dielectric breakdown strength. It is unlikely that the maximum polarization is controlled by the effect of high entropy.

(3) It is not clear why the P-E hysteresis hardly changes when the temperature is changed. With increasing temperature, the energy density is expected to gradually decrease due to a decrease in dielectric constant and breakdown strength (decrease in insulation resistance). Why is the energy density nearly constant over the temperature range from room temperature to 150°C?

Reviewer #3 (Remarks to the Author):

This manuscript describes a new energy storage dielectric with tetragonal tungsten bronze structure. Specifically, excellent energy density and performance stability are achieved through disorder induced by element doping and improvement in insulation. After carefully considering the content of the manuscript, I make the following comments to urge further improvement of the quality of the manuscript.

1. I do not believe that changes in TTB dielectric and relaxation behavior are directly related to compositional disorder, such as that observed in Ba₅La₂Zr₃Nb₇O₃₀ and Ba₅Sm₂Zr₃Nb₇O₃₀, unlike perovskites. The latter exhibits a pseudocubic structure and a dielectric constant peak that changes with frequency in sufficient disorder.
2. Although the ionic radii of Ta and Nb are similar, the impact on the unit cell volume is usually different due to differences in force constants.
3. What causes CaNb₂O₆ and/or Ba₆Nb₂O₁₁ to appear? By the way, uncontrollable secondary phases are usually detrimental to industry.
4. For the observation of PNRs, perovskites are generally observed along [001]-axis, which is beneficial for determining the direction and local symmetry of the polarization vector. Why did the authors choose the [110]-axis as the observation direction?
5. Generally speaking, the conductance activation energy is half of the intrinsic band gap. Why does the band gap of BSCNT_{0.45} increase under the action of the second phase, but the conductance activation energy decreases instead?
6. Why is no obvious phase transition phenomenon observed in the in-situ XRD results? This is different from common perovskites such as BNT.
7. For practical applications, the evaluation of the discharge capability of ceramic capacitors is very important. I suggest the authors to supplement relevant experimental results,

including discharge speed and temperature stability. In addition, the detailed parameters of the discharge circuit should also be given, since the discharge behavior critically depends on the construction of the circuit.

Response Letter to Reviewers

We appreciate the reviewers for their efforts and time spent on our manuscript. The comments are highly valuable and extremely helpful for us to further improve the quality of our manuscript. Now we have made all necessary revisions according to the reviewers' comments. In the following, our detailed responses to the reviewers' comments are given in a point-to-point manner. Note that all the corresponding modifications are highlighted in our revised manuscripts. **Other imperfections and typos in English have also been amended. In addition, we added the **Methods** section into the manuscript according to the standard format requirements of Nature Communications.**

Reviewer #1

The manuscript describes strategies to improve energy storage performance in BSCNT ceramics based on high entropy and band gap engineering, and demonstrates its powerful advantages to readers. The performance parameters are attractive enough, whether it is perovskite or tungsten bronze. The experimental data are abundant and sufficient to support most of the authors' conclusions, however, there are still confusing aspects of the scientific explanation. I, therefore, recommend that the manuscript be significantly revised by the authors and reviewed again for consideration for publication by Nature Communications.

Response: Thanks for your positive comments. In the following, we have provided point-to-point responses to the comments and suggestions. All modified parts have been **highlighted**.

Comment 1. The important description of previous results is omitted in the introduction. The structural and electrical behavior of $\text{Sr}_x\text{Ba}_{1-x}\text{Nb}_2\text{O}_6$ bronze has been studied for many years, with important contributions from institutions including the National Institute of Standards and Technology and Penn State University. In this regard, the authors should enrich the overview of SBN structural complexity (as stated by the authors in Page 4, lines 62-63) and basic dielectric properties. Please refer to 10.1063/1.1657277 and 10.1021/acs.chemmater.9b04122, etc.

Response 1. Thanks for the good comments. Based on the suggestions, we have carefully studied previous reports on the structure and dielectric properties of SBN, especially the provided references, and have now summarized as follows:

Detailed modifications in the manuscript: (see pages 5 and 6 in the revised manuscript)

$\text{Sr}_x\text{Ba}_{1-x}\text{Nb}_2\text{O}_6$ (SBN) is an actively studied composition featuring a tetragonal tungsten bronze structure (TTBs). Unlike filled structures, SBN is classified as an unfilled TTBs due to the presence of a 1/6 vacancy at the A site, which is randomly distributed in the A1 and A2 sites. Typically, the larger Ba^{2+} ions exclusively occupy the A2-site, while the smaller Sr^{2+} ions are distributed among both A1 and A2 sites¹. The widely accepted explanation for the relaxation behavior in TTB system involves lattice distortion resulting from the randomness of cationic arrangements. The relaxation behavior in SBN, particularly in $\text{Sr}_{0.67}\text{Ba}_{0.33}\text{Nb}_2\text{O}_6$ material, is attributed to the size mismatch between Sr and Ba ions at A2 sites, displacing adjacent oxygen atoms and thereby inducing a net polarization effect². Additionally, the specific ion

occupancy in the crystal structure reveals that the ferroelectric relaxation phase transition in SBN occurs when more than half of the Sr^{2+} occupy the A2-site. The transition from normal ferroelectric to relaxor state is gradual, occurring for compositions with $x \geq 0.60$ in single crystals and $x \geq 0.53$ in ceramics³.

Comment 2. Page 11, lines 190-191. The authors claim that "...the Sr^{2+} that can occupy both A1 and A2, the disparity between A1 and A2 positions increases, leading to further lattice inhomogeneity." The specific meaning of "lattice inhomogeneity" here should be pointed out. In addition, why do differences in lattice site size affect "lattice inhomogeneity"?

Response 2. Thanks for the good comments. We apologize for the inaccurate explanation of "lattice inhomogeneity". We actually intend to convey that the addition of Ca^{2+} significantly alters the distribution of different cations in TTBs, particularly increasing the disparity between A1 and A2. According to the reviewer's feedback, we have revised this sentence to provide a clearer description of the result. In the case of $\text{Ba}_{0.4}\text{Sr}_{0.6-x}\text{Ca}_x\text{Nb}_{2-x}\text{Ta}_x\text{O}_6$ ceramics, the larger Ba^{2+} ion (with a radius of 1.61 Å) exclusively occupies the A2 sites, while the smaller Sr^{2+} ion (with a radius of 1.44 Å) is distributed among both A1 and A2 sites. The Ca^{2+} ion (with a radius of 1.34 Å), on the other hand, can only occupy the A1-site due to its smaller cationic radius⁴.

In order to further substantiate our argument, we have performed additional Rietveld refinement analysis on the XRD results. Figure R1 illustrates the refinement results of XRD for BSCNT0 and BSCNT0.30, with corresponding cationic positions

and proportions provided in Tables R1 and R2. In BSCNT0, the proportions of A2-sites occupied by Sr^{2+} and Ba^{2+} are 38.21% and 51.52%, respectively. A1-sites are solely occupied by Sr^{2+} at a proportion of 70.8%, with the remainder being vacant. In BSCNT0.30 ceramics, Ca^{2+} occupies 70.5% of the A1-sites, while Sr^{2+} occupies 19.2%. Simultaneously, A2-sites are occupied by 52.5% of Ba^{2+} and 23.0% of Sr^{2+} , with the remaining sites vacant. Based on the provided data, the significant difference in the proportion occupied by different cations at A1 and A2-sites can be observed, and the following results can be calculated: without accounting for the vacancy, the mean ion radius of BSCNT0 at the A1-sites is 1.44 Å, and at the A2-sites is 1.54 Å; for BSCNT0.30, the mean ionic radius is 1.36 Å at the A1-sites and 1.55 Å at the A2-sites. This difference will cause lattice distortion, thereby affecting the polarization and corresponding energy storage performance.

Figure R1 (Figure S2 in the revised Supporting Information) Rietveld refinement results of

(a) BSCNT0 and (b) BSCNT0.30.

Table R1 (Table S1 in the revised Supporting Information) The cationic position and proportion of BSCNT0 ceramics obtained XRD refinement results

Name	Fractional coordinates			Occupancy
Ba2	0.172144	0.672144	0.503508	0.5152

Sr2	0.172738	0.672738	0.493875	0.3821
Sr1	0.000000	0.000000	0.489268	0.7082
Nb1	0.500000	0.000000	0.015835	0.9987
Nb2	0.074493	0.211493	0.001110	0.9971
O1	0.282125	0.782125	0.968522	1.0628
O2	0.138593	0.068118	0.958397	0.9683
O3	0.993724	0.341702	0.958619	1.0299
O4	0.500000	0.000000	0.478860	1.0045
O5	0.074595	0.205913	0.466276	1.0154

Table R2 (Table S2 in the revised Supporting Information) The cationic position and proportion of BSCNT0.30 ceramics obtained XRD refinement results

Name	Fractional coordinates			Occupancy
Ba2	0.172491	0.672491	0.516609	0.5253
Sr2	0.172189	0.672190	0.510159	0.2802
Sr1	0.000000	0.000000	0.498482	0.1923
Ca1	0.000000	0.000000	0.490252	0.7053
Nb1	0.500000	0.000000	0.020352	0.8494
Ta1	0.500000	0.000000	0.020352	0.1533
Nb2	0.074552	0.211098	0.009753	0.8494
Ta2	0.074552	0.211098	0.009753	0.1533
O1	0.282076	0.782076	0.949705	1.0298
O2	0.140983	0.069492	0.948012	1.0371
O3	0.991005	0.338295	0.937655	0.9979
O4	0.500000	0.000000	0.525831	0.9737
O5	0.072395	0.205210	0.509134	1.0464

Detailed modifications in the manuscript: (see pages 12 in the revised manuscript)

The Rietveld refinement results of XRD data for BSCNT0 and BSCNT0.30 are illustrated in Figure S2. The mean ionic radii of the distinct A-sites within the respective components were computed based on the cationic position and proportion (Tables S1 and S2). The results showcase that the mean ionic radius of BSCNT0 is 1.44 Å and 1.54 Å at A1-sites and A2-sites respectively. For BSCNT0.30, on the other hand, the mean ionic radius is 1.36 Å at the A1-sites and 1.55 Å at the A2-sites. The augmented

disparity in radius between A1 and A2 sites is expected to induce lattice distortion, which will affect the polarization and corresponding energy storage performance.

Detailed modifications in the Supporting Information: (see pages 3 and 4 in the revised Supporting Information)

Figure S2 The Rietveld refinement results of XRD data for (a) BSCNT0 and (b) BSCNT0.30.

Table S1 The cationic position and proportion of BSCNT0 ceramics obtained XRD refinement results

Name	Fractional coordinates			Occupancy
Ba2	0.172144	0.672144	0.503508	0.5152
Sr2	0.172738	0.672738	0.493875	0.3821
Sr1	0.000000	0.000000	0.489268	0.7082
Nb1	0.500000	0.000000	0.015835	0.9987
Nb2	0.074493	0.211493	0.001110	0.9971
O1	0.282125	0.782125	0.968522	1.0628
O2	0.138593	0.068118	0.958397	0.9683
O3	0.993724	0.341702	0.958619	1.0299
O4	0.500000	0.000000	0.478860	1.0045
O5	0.074595	0.205913	0.466276	1.0154

Table S2 The cationic position and proportion of BSCNT0.30 ceramics obtained XRD refinement results

Name	Fractional coordinates			Occupancy
Ba2	0.172491	0.672491	0.516609	0.5253
Sr2	0.172189	0.672190	0.510159	0.2802
Sr1	0.000000	0.000000	0.498482	0.1923
Ca1	0.000000	0.000000	0.490252	0.7053
Nb1	0.500000	0.000000	0.020352	0.8494

Ta1	0.500000	0.000000	0.020352	0.1533
Nb2	0.074552	0.211098	0.009753	0.8494
Ta2	0.074552	0.211098	0.009753	0.1533
O1	0.282076	0.782076	0.949705	1.0298
O2	0.140983	0.069492	0.948012	1.0371
O3	0.991005	0.338295	0.937655	0.9979
O4	0.500000	0.000000	0.525831	0.9737
O5	0.072395	0.205210	0.509134	1.0464

Comment 3. Page 13, lines 229-231. The statement “This suppression of grain growth is attributed to the sluggish diffusion effect observed in high-entropy materials” needs to be corrected. In fact, Ca or Ta are very common as grain growth inhibitors. Experimental phenomena, therefore, cannot be attributed solely to changes in entropy.

Response 3. Thanks for the good comments. We agree with the reviewer that the high refractory behavior and high melting point of Ca and Ta oxides enable them to effectively suppress grain growth, consequently leading to an increase in the breakdown field of energy storage ceramics. We have emphasized this important conclusion in the revised manuscript as suggested. Additionally, we believe that the reduced grain size is also influenced by the slow diffusion effect resulting from the high entropy effect. A comprehensive understanding of high-entropy materials has been proposed through the identification of the “four core effects” - high-entropy, severe-lattice-distortion, sluggish-diffusion, and cocktail effects - which are commonly employed to elucidate the mechanisms underlying various unique phenomena associated with high-entropy materials ⁵. As the configuration entropy increases, the lattice gradually distorts, significantly impacting the diffusion of each atom. In a heavily distorted lattice, the diffusion rate of species is notably reduced compared to pristine lattices with equivalent average bond strength and melting temperature. This is due to the increased difficulty

for atoms or vacancies to move along their jumping paths in lattice sites, consequently elevating the diffusion activation energy. This substantial consequence of the sluggish-diffusion effect in severely distorted lattices has numerous advantages, with inhibiting grain growth being one of the most significant ⁶. This will be beneficial to improve the breakdown field thus the energy storage density.

Detailed modifications in the manuscript: (see pages 14 and 15 in the revised manuscript)

Given their remarkable refractory behavior, oxides of Ta and Ca are extensively employed as grain growth inhibitors to mitigate the average grain size ^{7, 8}. Furthermore, the lattice distortion induced by the elevated configurational entropy can substantially impede atomic diffusion, thereby further diminishing the average grain size thus higher grain boundary density ⁶. Notably, grain boundaries with higher resistivity play an important role in preventing electric breakdown by absorbing the energy from electric tree expansion within the grains ⁹.

Comment 4. Page 14, lines 250-252, and Page 15, lines 269-271. Without giving the specific properties of the impurity phase, it is believed that its damage to performance is flawed.

Response 4. Thanks for the valuable comment. We have revised the relevant expression in the paper, analyzed the properties of impurity phase and its impact on performance.

Detailed modifications in the manuscript: (see pages 16 and 17 in the revised manuscript)

Nevertheless, in the BSCNT0.45 ceramics, the solid solubility limitation of Ca^{2+} ions within the SBN matrix gives rise to secondary phases, such as $\text{Ca}_2\text{Nb}_2\text{O}_6$ and/or $\text{Ba}_6\text{Nb}_2\text{O}_{11}$. Notably, oxygen vacancy exists in $\text{Ba}_6\text{Nb}_2\text{O}_{11}$ to maintain electrical neutrality¹⁰. Since the activation energy inversely correlates with the concentration of oxygen vacancies¹¹, BSCNT0.45 exhibits a lower activation energy, which is further supported by the decreased breakdown field observed in BSCNT0.45.

Comment 5. Page 18, lines 323-325. The presence of incommensurability may not necessarily be related to relaxation behavior, as is observed in antiferroelectrics (10.1103/PhysRevB.72.024102), but simply exists as a structural feature that may be related to ion displacements. In addition, the authors ignored the impact of B site disorder (as mentioned by the authors in the previous section) on δ .

Response 5. Thanks for the valuable comments. We totally agree with the reviewer that the presence of incommensurability may not necessarily be related to relaxation behavior. However, for TTBs materials, numerous studies have shown that the incommensurate structure is related to the relaxation property, with the incommensurability parameters positively correlated with the relaxation property^{12 13, 14, 15, 16}.

A modulation here is defined as a periodic deformation of a “basic structure” with space-group symmetry. If the periodicity of the modulation does not align with the periodicities of the basic structure, the modulated crystal structure is termed as incommensurate. This description is based on the observation in the diffraction pattern of modulated structures, where the main reflections are situated on a reciprocal lattice,

and additional reflections, generally of weaker intensity, are referred to as satellites. The modulation that gives rise to the satellites can be a displacive modulation, involving a periodic displacement from the atomic positions of the basic structure, or an occupation modulation, in which the atomic positions of the basic structure are occupied with a periodic probability function. Mixed forms also occur. The presented modulated crystal case is the simplest one giving rise to an incommensurate crystal structure.

For the antiferroelectric $\text{Pb}_{0.99}\text{Nb}_{0.02}(\text{Zr}_{1-x}\text{Sn}_x)_{1-y}\text{Ti}_y\text{O}_3$ (10.1103/PhysRevB.72.024102) as referenced by the reviewer, the incommensurate modulation exists in the form of a transverse Pb-cation displacement wave. These modifiers (Nb^{5+} , Sn^{4+} and Ti^{4+}) introduce unique nanoscale structural feature to the ceramics in the form of incommensurate modulations. The observed incommensurate modulation at large scale is suggested to be a mixture of commensurate modulations at local scale.

The occurrence of incommensurate modulation can be attributed to various factors. However, in the case of TTBs, several studies have indicated that commensurate modulation is associated with ferroelectricity, whereas incommensurate modulation is linked to relaxation transition^{12 13}. It is believed that the increase in ionic disorder is the cause of the appearance of incommensurately modulated satellite diffraction spots, and the greater the disorder, the higher the value of the incommensurability parameter δ , aligning with the mechanism of enhanced relaxation properties¹⁷. The microscopic origins of PNRs are closely tied to the inherent structural and charge inhomogeneities

which are typical for relaxors. It is crucial to consider the commensurate to incommensurate transition as essentially a change in tilt configuration of the oxygen sublattice¹³.

To more clearly demonstrate the correlation between incommensurate modulation and relaxation characteristics, and to highlight the positive relationship between the incommensurate parameter values and relaxation properties, we have summarized the data for various TTBs dielectrics in Table R3. In this table, compounds within the same group are ordered according to their relaxation characteristics, ranging from weak to strong. This arrangement underscores the progressive enhancement of relaxation properties with increasing incommensurability parameters.

Table R3 A summary of relaxation characteristics and incommensurability parameter of some actively studied TTBs dielectrics

Number	Compound	δ (incommensurability coefficient)	References	Ferroelectric(FE)/Relaxor(RE)
1	$\text{Sr}_2\text{NaNb}_5\text{O}_{15}$	0 (commensurate)	18	FE
	$\text{Sr}_2\text{Na}_{0.85}\text{Bi}_{0.05}\text{Nb}_5\text{O}_{15}$	0.0028	19	RE
	$\text{Sr}_2\text{Na}_{0.85}\text{Bi}_{0.05}\text{Nb}_{4.4}\text{Ta}_{0.6}\text{O}_{15}$	0.0079	19	RE
	$\text{Sr}_2\text{Na}_{0.85}\text{Bi}_{0.05}\text{Nb}_4\text{Ta}_1\text{O}_{15}$	0.0138	19	RE
2	$\text{Ca}_{0.4}\text{Sm}_{0.2}\text{Ba}_{1.8}\text{Nb}_5\text{O}_{15}$	0.078	14	RE
	$\text{Ca}_{0.4}\text{Bi}_{0.2}\text{Ba}_{1.8}\text{Nb}_5\text{O}_{15}$	0.094	14	RE
3	$\text{Gd}_{0.03}\text{Ba}_{0.47}\text{Sr}_{0.485}\text{Nb}_2\text{O}_6$	0.151 ± 0.01	15	RE
	$\text{Gd}_{0.03}\text{Ba}_{0.47}\text{Sr}_{0.455}\text{Sm}_{0.02}\text{Nb}_2\text{O}_6$	0.283 ± 0.06	15	RE
4	$\text{Ba}_2\text{SmTi}_2\text{Nb}_3\text{O}_{15}$	commensurate	16	FE
	$\text{Ba}_2\text{GdTi}_2\text{Nb}_3\text{O}_{15}$	commensurate	16	FE
	$\text{Ba}_2\text{NdTi}_2\text{Nb}_3\text{O}_{15}$	commensurate	16	FE
	$\text{Ba}_2\text{BiTi}_2\text{Nb}_3\text{O}_{15}$	>0	16	RE
	$\text{Ba}_2\text{LaTi}_2\text{Nb}_3\text{O}_{15}$	0.04	16	RE
5	$\text{Ba}_{0.4}\text{Sr}_{0.6}\text{Nb}_2\text{O}_6$	0.18 ± 0.04	This work	RE
	$\text{Ba}_{0.29}\text{Sr}_{0.71}\text{Nb}_2\text{O}_6$	0.26 ± 0.05	20	RE
6	$\text{Ba}_{0.4}\text{Sr}_{0.6}\text{Nb}_2\text{O}_6$	0.18 ± 0.04	This work	RE
	$\text{Ba}_{0.4}\text{Sr}_{0.3}\text{Ca}_{0.3}\text{Nb}_{1.70}\text{Ta}_{0.3}\text{O}_6$	0.28 ± 0.06	This work	RE

Based on the preceding discussion, it can be inferred that the introduction of Ta⁵⁺ at the B-site contributes to the heightened disorder among the ions at the B site, thereby amplifying lattice distortion and deformation¹⁹. This phenomenon aligns with the prerequisites for intensified incommensurate modulation and warrants further exploration. We have included related discussions in the manuscript.

At the same time, it has come to our attention that there was a clerical error in the representation of $\delta_{\text{BSCNT}0.30}$, which was mistakenly recorded as 0.48 ± 0.06 instead of the correct data, which is 0.28 ± 0.06 . We sincerely apologize for this error. It is important to note, however, that this clerical error does not affect the variability trend of δ and supports the original conclusion.

Detailed modifications in the manuscript: (see page 20 in the revised manuscript)

Simultaneously, the addition of Ta⁵⁺ causes unavoidable distortion of the oxygen octahedron, leading to local lattice distortion. This suggests that B-site ion doping tends to promote incommensurate modulation¹⁹.

Comment 6. The most important issue is that the manuscript appears to only demonstrate the impact of bandgap engineering on energy storage performance. I carefully checked the authors and literature in similar fields, and found that the change in the dielectric behavior of BSCNT is very similar to the doping of a single element (10.1002/adma.202310559). The authors should further emphasize the important role of high entropy. Furthermore, classifying BSCNT as the high-entropy ceramics seems a bit reluctant, especially without giving detailed structural parameters.

Response 6. Thanks for the good comment. Both GBSSN and BSCNT ceramics

are SBN-based unfilled TTB structured ceramics. The ultimate goal is to enhance relaxation characteristics through element doping while increasing breakdown fields to improve energy storage performance. So it is reasonable that the dielectric behavior of the two is similar. But there are still some differences between the two types of ceramics. For convenience of comparison, we compared the dielectric constant of single element doping (10.1002/adma.202310559) with that of this work (to be more intuitive, we changed the temperature range of the dielectric temperature spectrum of this work), as shown in Figure R2. Since the doping amount of GBSSN is very small, the decrease in dielectric constant is not obvious. However, in this work, a relatively large amount of doping is required to meet the requirement of increasing configuration entropy, so the decrease in dielectric constant is more obvious. T_m also moves further downwards to lower temperature.

High-entropy ferroelectrics not only display significant compositional disorder but also exhibit a disordered distribution of ferroelectric distortion (polarization vectors and/or oxygen octahedral tilts) at local scale²¹. Finally, it exhibits significant relaxation properties, as evidenced by the smeared dielectric peaks and pronounced frequency dispersion observed in the temperature dependence of dielectric constant and loss tangent. To illustrate the above information more directly, the dielectric behavior between GBSSN and BSCNT is compared in Table R4.

High-entropy ceramics are characterized by a configurational entropy value $\Delta S_{\text{config}} \geq 1.5R$, qualifying them as “high entropy”. In this work, we calculated the configurational entropy of individual components, as depicted in Fig. 1(a). The

calculated results are as followings: BSCNT0 (0.67R), BSCNT0.15 (1.28R), BSCNT0.30 (1.51R), and BSCNT0.45 (1.54R). By definition, both BSCNT0.30 and BSCNT0.45 are high entropy ceramics. As the configuration entropy increases, the lattice gradually distorts, significantly impacting the diffusion of the ions. In a heavily distorted lattice, the diffusion rate of ions is notably reduced compared to pristine lattices with equivalent average bond strength and melting temperature. This is due to the increased difficulty for ions or vacancies to move along their jumping paths in lattice sites, consequently elevating the diffusion activation energy. One of the most significant consequences of the sluggish-diffusion effect in severely distorted lattices is the inhibition of grain growth⁵. Meanwhile, the introduction of multiple elements into equivalent lattice positions via the high entropy strategy increases chemical complexity, disrupts long-range order in the ferroelectrics, and enhances the relaxor characteristics of BSCNT0.30²², as illustrated in Fig. 4.

In the revised manuscript, we underscored the impact of high entropy on the breakdown field and elucidate how the high entropy strategy affects relaxation properties. Our aim is to optimize energy storage performance by adjusting elemental composition to increase configurational entropy. However, it is important to note that increasing configurational entropy requires doping with multiple elements or adjusting the proportions of different elements, which inevitably affects the performance and structure of the sample due to the dopant characteristics. Therefore, we also acknowledged the significance of doping element properties in shaping energy storage performance in the revised manuscript.

Figure R2 (a) Temperature dependence of the dielectric constant and loss tangent ($\tan\delta$) of the GBSSN (10.1002/adma.202310559). (b) Temperature dependence of the dielectric constant and loss tangent ($\tan\delta$) of the BSCNT ceramics at 1 kHz (This work).

Table R4 Comparison of dielectric behavior between GBSSN (10.1002/adma.202310559) and

BSCNT (This work)

GBSSN	$T_m(1kHz)$	$\epsilon_m(1kHz)$	$\Delta T_m(T_m@10Hz - T_m@2MHz)$	γ	BSCNT	$T_m(1kHz)$	$\epsilon_m(1kHz)$	$\Delta T_m(T_m@10Hz - T_m@2MHz)$	γ
$x=0$	45.1	2705	29.5	1.53	$x=0$	58	3223	17	1.35
$x=0.01$	33.8	2341	36.8	1.57	$x=0.15$	27.8	2275	41.1	1.54
$x=0.02$	22.6	2259	41.9	1.62	$x=0.30$	-3	1472	51.8	1.73
$x=0.03$	11.1	2000	45.7	1.67	$x=0.45$	-57.4	768	59	1.68

Detailed modifications in the manuscript: (see page 18 in the revised manuscript)

The introduction of multiple elements into equivalent lattice positions via the high entropy strategy increases chemical complexity, disrupts long-range ordering in the ferroelectrics, and enhances the relaxor characteristics of BSCNT0.30²².

Reviewer #2

This paper shows that $\text{Ba}_{0.4}\text{Sr}_{0.3}\text{Ca}_{0.3}\text{Nb}_{1.7}\text{Ta}_{0.3}\text{O}_6$ with a tetragonal tungsten bronze-type structure has a high energy density and is an excellent dielectric material for energy applications.

The superiority of this material over other lead-free materials in terms of energy density, energy storage efficiency, and the temperature dependence should be recognized as engineering, but the chemical composition system of the material is not new. I think that novel concepts and design guidelines that are generally applicable to tungsten bronze-type compounds are needed to be published in Nature commun. The authors explain various phenomena as effects of high entropy, assuming that high entropy is important for obtaining high energy density, but some of the explanations are not satisfactory.

Response: Thanks for the positive comments. In the following, we have provided point-to-point responses to the comments and suggestions. All modified parts have been highlighted.

Comment 1. In this study, Sr and Nb are partially replaced by Ca and Ta in order to increase the entropy. The relaxor-like change can certainly be explained by the entropy increase, but there is no evidence that the dielectric breakdown strength is an effect of high entropy. In the case of perovskite-type oxides, the dielectric breakdown strength is more likely to be an elemental effect derived from Ca and Ta, as CaTiO_3 and KTaO_3 have higher dielectric strength than SrTiO_3 and KNbO_3 , respectively. There is also no evidence that the smaller grain size is also due to a high entropy effect; the

introduction of Ca and Ta may simply increase the melting point.

Response 1. Thanks for the good comments. We agree with the reviewer that the increased breakdown strength is predominantly influenced by the remarkable refractory behavior and large band gap of Ta₂O₅ and CaO^{7,8}. Furthermore, the lattice distortion induced by elevated configurational entropy can substantially impede atomic diffusion, thereby further diminishing the average grain size. This sluggish-diffusion effect enhances the breakdown field by inhibiting grain growth. We have included pertinent details to elucidate this aspect.

Detailed modifications in the manuscript: (see pages 14 and 15 in the revised manuscript)

Given their remarkable refractory behavior, oxides of Ta and Ca are extensively employed as grain growth inhibitors to mitigate the average grain size^{7,8}. Furthermore, the lattice distortion induced by the elevated configurational entropy can substantially impede atomic diffusion, thereby further diminishing the average grain size thus higher grain boundary density⁶. Notably, grain boundaries with higher resistivity play an important role in preventing electric breakdown by absorbing the energy from electric tree expansion within the grains⁹.

Comment 2. The maximum polarization value of Ba_{0.4}Sr_{0.3}Ca_{0.3}Nb_{1.7}Ta_{0.3}O₆ is not very large for a tetragonal tungsten bronze-type oxide, and the high energy density is mostly due to the high dielectric breakdown strength. It is unlikely that the maximum polarization is controlled by the effect of high entropy.

Response 2. We apologize for any confusion caused by our previous statement.

We agree with the reviewer that the maximum polarization did not increase. In fact, the maximum polarization of BSCNT0.30 decreased compared with BSCNT0 at the same electric field, due to the decrease of dielectric constant, as shown in Figure S1. However, due to the obvious increase of the breakdown strength, BSCNT0.30 obtains a higher P_{\max} than BSCNT0 under the critical breakdown field. This result is due to the delay of polarization saturation caused by relaxation characteristics, so that the polarization continues to gradually increase with the applied electric field at a higher electric field, and the increase of the superimposed breakdown field leads to the increase in the final P_{\max} . This is also one of the important advantages of relaxation ferroelectrics applied to energy storage. According to this comment, we have removed the relevant misleading expression.

Detailed modifications in the manuscript: (see page 22 in the revised manuscript)

The rapid reversibility of PNRs leads to reduced P_r , ~~higher P_{\max}~~ delayed polarization saturation, and ultimately exceptional energy storage performance

Comment 3. It is not clear why the P-E hysteresis hardly changes when the temperature is changed. With increasing temperature, the energy density is expected to gradually decrease due to a decrease in dielectric constant and breakdown strength (decrease in insulation resistance). Why is the energy density nearly constant over the temperature range from room temperature to 150°C?

Response 3. Thanks for the good comments. The *P-E* loops, reflecting the relationship between polarization and applied electric field in ferroelectric materials, is

intricately linked to factors such as crystal structure, lattice distortion, and atomic arrangement within materials. Temperature fluctuations can impact the crystal structure and atomic arrangement of materials, thereby influencing their ferroelectric properties.

In this study, the excellent temperature stability of BSCNT0.30's energy storage performance is attributed to the temperature insensitivity of its crystal structure (as shown in Figures 4c and 4d) and the ultra-low dielectric loss of less than 0.003 in the range of room temperature to 150°C. Moreover, the complex impedance spectrum (Figure S4 (c)) indicates that BSCNT0.30 maintains extremely high resistance (~2MΩ) even at 450°C, demonstrating its excellent high-temperature insulation properties. We agree with the reviewer that both dielectric constant ($T_m \sim -16^\circ\text{C}$) and breakdown strength will decrease with increasing temperature. For BSCNT0.30 ceramic, the dielectric constant variation across the temperature range up to 180°C is relatively small, remaining below 49%. Meanwhile, the applied electric field during temperature variation test was set at 500 kV/cm, significantly lower than its room temperature breakdown field (700 kV/cm). To analyze the temperature dependence of the P-E loop of BSCNT0.30 in more detail, Figure R3 summarizes P_{\max} and P_r in the range of 25°C to 180 °C. As shown in the figure, both P_r and P_{\max} exhibit similar variation trend with increasing temperature, accounting for a similar integrated areas in the PE plots at different temperatures. At 180°C, the increase in P_{\max} is attributed to additional charge from increased leakage at high temperatures, while the increase in P_r may be related to conduction losses due to thermal stimulation, ultimately leading to lower efficiency²³.

Additionally, the excellent temperature stability may be attributed to the superparaelectric (SPE) state of BSCNT0.30 within the temperature range of RT to 180°C. In SPEs, nanodomains are further reduced in size, and domain intercoupling is weakened to the extent that domain switching energy barriers become comparable to or lower than the thermal disturbance energy kT (where k is the Boltzmann constant). This allows the polarization of nanodomains to switch among energy-equivalent directions with high dynamics, enabling minimal hysteresis. The temperature-driven SPE in relaxors was reported to maintain nonlinear polarization with high P_{\max} and minimal hysteresis, which is desirable for achieving both high W_{rec} and η in dielectric capacitors.

Figure R3. Temperature dependence of the P_r and P_{\max} of BSCNT0.30.

Based on the reviewer’s feedback, we have added the following explanation in the manuscript regarding the temperature dependent energy storage properties.

Detailed modifications in the manuscript: (see pages 23 and 24 in the revised manuscript)

This further validates the excellent temperature stability of the BSCNT0.30 ceramics, indicating their crystal structure is insensitive to temperature variation. The

excellent temperature stability of BSCNT0.30's energy storage performance is attributed to the temperature insensitivity of its crystal structure (as shown in Figures 5(c) and 5(d)) and the ultra-low dielectric loss of less than 0.003 across the temperature range up to 180°C. Moreover, the complex impedance spectrum indicates that BSCNT0.30 maintains extremely high resistance ($\sim 2\text{M}\Omega$) even at 450°C, demonstrating its excellent high-temperature insulation properties. The excellent temperature stability may also be attributed to the superparaelectric (SPE) state of BSCNT0.30. The temperature-driven SPE in relaxors was reported to maintain nonlinear polarization with high P_{max} and minimal hysteresis, which is desirable for achieving both high W_{rec} and η in dielectric capacitors²⁴. At 180°C, the increase in P_{max} is attributed to additional charge from increased leakage at high temperatures, while the increase in P_r may result from conduction losses due to thermal stimulation, ultimately leading to lower efficiency²³.

Reviewer #3:

This manuscript describes a new energy storage dielectric with tetragonal tungsten bronze structure. Specifically, excellent energy density and performance stability are achieved through disorder induced by element doping and improvement in insulation. After carefully considering the content of the manuscript, I make the following comments to urge further improvement of the quality of the manuscript.

Response: Thanks for the very positive comments. In the following, we have provided point-by-point responses. All revisions have been highlighted in the manuscript.

Comment 1. I do not believe that changes in TTB dielectric and relaxation behavior are directly related to compositional disorder, such as that observed in $\text{Ba}_5\text{La}_2\text{Zr}_3\text{Nb}_7\text{O}_{30}$ and $\text{Ba}_5\text{Sm}_2\text{Zr}_3\text{Nb}_7\text{O}_{30}$, unlike perovskites. The latter exhibits a pseudocubic structure and a dielectric constant peak that changes with frequency in sufficient disorder.

Response 1. Thanks for the good comment. The relaxation properties of relaxation materials are attributed to their unique polarization structure, specifically the presence of polar nanoscale regions (PNRs). The microscopic origins of PNRs are closely linked to the inherent structural and charge inhomogeneities.

The crossover to the relaxation behavior in $\text{Sr}_x\text{Ba}_{1-x}\text{Nb}_2\text{O}_6$ with larger x is considered to be related to an enhancement of structural and related charge disorders. These disorders are primarily due to the randomly distributed vacancies on A-sites²⁵. The vacancies on these sites are the primary sources of random electric fields due to

the missing charges. Additionally, the presence of vacancies on both A1 and A2-sites further enhances this disorder. The accommodation misfits of the different oxygen octahedra lead to local buckling and tilting deformation, creating localized electric multipole moments, which also contribute to the random electric fields²⁵.

From a statistical perspective, the most ordered structure is anticipated in $\text{Sr}_{0.2}\text{Ba}_{0.8}\text{Nb}_2\text{O}_6$, where all A2-sites are exclusively occupied by Ba^{2+} cations, while the Sr^{2+} ions and vacancies are distributed on the A1-sites. As the Sr/Ba molar ratio increases, a transition from ferroelectric to relaxor behavior is observed in compositions with $x > 0.6$ ²⁵. Additionally, it has been suggested that vacancies on A2-sites have a more pronounced impact on the polar properties of SBN compared to those at A1-sites²⁶. As the vacancy tends to occupy the A2-sublattice more frequently with increasing x , the corresponding “active” charge disorder is heightened with a higher Sr^{2+} content²⁶. In summary, in TTBs ceramics with SBN-based unfilled structure, component disorder is a significant factor inducing relaxation behavior.

Furthermore, for filled TTBs, the relaxation properties also display unique characteristics due to the absence of the A-sites vacancy. The relaxation characteristics are primarily determined by the radius of ions in the A-site¹³. For example, in the $\text{Ba}_4\text{R}_2\text{Zr}_4\text{Nb}_6\text{O}_{30}$ (R = La, Nd, Sm) filled TTBs system, the transition from ferroelectric to relaxor behavior is observed as the R ion radius increases. The author posits that the radius difference of the A-site ions primarily governs this transformation. Simultaneously, low-temperature dielectric relaxation and ferroelectric microdomains were observed in all three components due to the increased B-site disorder and strong

lattice distortion caused by the addition of Zr²⁷. For Ba₅RZr₃Nb₇O₃₀ (R = La, Nd, Sm), on the other hand, despite all three compounds show relaxation behavior, when the rare earth ion radius at the A1 site decreases, the dispersion of the phase transition decreases and the relaxation characteristics gradually weaken²⁸.

Despite the number of elements in the system remaining constant, the varying structural distortions resulting from different rare earth element radii ultimately lead to changes in relaxation behavior. This relaxation behavior, dominated by structural inhomogeneity, aligns with the origin of the relaxation properties discussed in this study. The fundamental approach to enhancing relaxation characteristics through the high entropy strategy (increasing the configuration entropy) in this study is to adjust the A-site occupancy and ionic radius difference (including vacancy distribution), while also increasing the variety of B-site elements to achieve the goal of inducing local lattice distortion. It is important to note that increasing element type or configuration entropy is not the sole method to enhance relaxation properties. Random fields can be introduced in various ways, such as adjusting the amount of charge to enhance local inhomogeneities, thus improving the relaxation properties.

Comment 2. Although the ionic radii of Ta and Nb are similar, the impact on the unit cell volume is usually different due to differences in force constants.

Response 2. Thanks for the valuable comments. We apologize for this mistake. We agree with the reviewer that Ta⁵⁺ and Nb⁵⁺ have numerous differences though they possess the similar ionic radii, these include electronegativity, polarizability, relative molecular mass, etc., where the electronegativity difference between metal cations and

oxygen ions determines the bond energy of B-O bonds. The electronegativity values for Ta⁵⁺, Nb⁵⁺, and O²⁺ are 1.5, 1.6, and 3.44, respectively. An increase in electronegativity difference may enhance the polarity in the chemical bond, leading to a larger shift of electrons in the chemical bond. Therefore, the addition of Ta⁵⁺ increases the polarity of the B-O bond, resulting in a reduction of the bond length and, consequently, the cell volume²⁹. According to this comment, we have revised the relevant content in the manuscript.

Detailed modifications in the manuscript: (see page 12 in the revised manuscript)

Moreover, despite the same ionic radius of Ta⁵⁺ (0.64 Å) and Nb⁵⁺ (0.64 Å), the increased electronegativity difference between Ta⁵⁺ and O²⁺ compared to Nb⁵⁺ leads to an increase in the polarity of the B-O bond. This, in turn, shortens the B-O covalent bond length, resulting in a contraction of the cell volume³⁰.

Comment 3. What causes CaNb₂O₆ and/or Ba₆Nb₂O₁₁ to appear? By the way, uncontrollable secondary phases are usually detrimental to industry.

Response 3. Thanks for the good comment. The secondary phase in BSCNT0.45 observed in this study is attributed to the specific distribution of metal cations with varying radii at the A-sites in the unfilled tetragonal tungsten bronze structure (TTBs). More precisely, the crystallographic configuration of the unfilled TTBs is represented by the formula A_{10.4-a}A_{20.8-b}B₂O₆, where a+b=0.2, indicative of the vacancies distributed at random within the A1 and A2 sites. In the context of Ba_{0.4}Sr_{0.6-x}Ca_xNb_{2-x}Ta_xO₆ ceramics, the larger Ba²⁺ ion, possessing an ionic radius of 1.61 Å, is found to

exclusively populate the A2 site. Conversely, the smaller Sr^{2+} ion, with an ionic radius of 1.44 Å, is distributed across both A1 and A2 sites. The Ca^{2+} ion, with a smaller ionic radius of 1.34 Å, is constrained to the A1 site. According to the stoichiometric formula, the A1 sites can theoretically accommodate a maximum of 0.4 Ca^{2+} ions, assuming that vacancies and Sr^{2+} ions are confined to the A2 sites exclusively ($a=0$, $b=0.2$). This scenario delineates the upper threshold of Ca^{2+} incorporation. Nonetheless, should the Ca^{2+} content increases further, additional phases are emerged. Empirically, due to the concurrent presence of Sr^{2+} ions and vacancies, the saturated solid solubility of Ca^{2+} is observed to be less than 0.4^{4, 31}, with the present of the secondary phase when x surpasses 0.3. Moreover, divergences in synthesis methodologies generate disparities in the maximal solid solution ratio of Ca^{2+} .

In this study, The BSCNT0.30 ceramics maintain a single tetragonal tungsten bronze structure, whereas BSCNT0.45 yields a secondary phase due to the aforementioned factors, which is detrimental to energy storage application. Following this comment, an elucidation of the mechanisms underpinning the formation of the secondary phase has been incorporated into the manuscript for comprehensive discussion.

Detailed modifications in the manuscript: (see page 11 in the revised manuscript)

In particular, one third of the A-site is occupied by A1-site, while two thirds are filled by A2-site. Additionally, in the unfilled structure, one sixth of the A-sites remain vacant and are randomly distributed between A1 and A2 sites. Therefore, even if all

vacancies and Sr^{2+} ions are located exclusively in the A2-site, the content of Ca^{2+} can only account for a maximum of one third of the A-site, corresponding to $x=0.4$. Beyond this range will result in the inability to maintain a single phase^{4, 31}.

Comment 4. For the observation of PNRs, perovskites are generally observed along [001]-axis, which is beneficial for determining the direction and local symmetry of the polarization vector. Why did the authors choose the [110]-axis as the observation direction?

Response 4. Thanks for the good comment. Given that perovskites demonstrate a pseudo-cubic structure, as illustrated in Fig. R4(a), it is observed that in the perovskite structure, all oxygen octahedra align either vertically or horizontally in the [001] orientation, as indicated by the blue arrows. This alignment renders this orientation favorable for observation. However, in the TTB structure, the oxygen octahedra in the [001] orientation are bent, as depicted by the blue arrows in Fig. R4(b). In this scenario, the polarization of atoms (shift) in this orientation may not be conducive to observation, as the periodicity of the lattice is not linear. The structure in the [110] orientation can address this issue, as depicted in Fig. R4(c), where all octahedra are aligned vertically and horizontally, making this orientation a better choice for determining local symmetry of the polarization vector in TTB structures.

Figure R4. The observed direction diagram of the polarization vector. (a) Perovskite structure in direction [001]; (b) TTB structure in direction [001]; (c) TTB structure in direction [110];

Comment 5. Generally speaking, the conductance activation energy is half of the intrinsic band gap. Why does the band gap of BSCNT0.45 increase under the action of the second phase, but the conductance activation energy decreases instead?

Response 5. Thanks for the good comment. In general, the conductance activation energy is half of the intrinsic band gap. Nevertheless, in the BSCNT0.45 ceramics, the solid solubility limitation of Ca^{2+} ions within the SBN matrix gives rise to the emergence of secondary phases, such as $\text{Ca}_2\text{Nb}_2\text{O}_6$ and/or $\text{Ba}_6\text{Nb}_2\text{O}_{11}$. Notably, $\text{Ba}_6\text{Nb}_2\text{O}_{11}$ exhibits an oxygen vacancy to maintain electrical neutrality¹⁰. Since the activation energy inversely correlates with the concentration of oxygen vacancies in dielectric ceramics, the BSCNT0.45 sample showcase an evidently lower activation energy¹¹. Additionally, the direct band gap measurement, conducted using a minute amount of ceramic powder, may not be sufficiently sensitive to trace impurities. Nevertheless, the measured direct band gap experiences an increase owing to the

elevated Ca and Ta content, contributing to a higher band gap.

Comment 6. Why is no obvious phase transition phenomenon observed in the in-situ XRD results? This is different from common perovskites such as BNT.

Response 6. Thanks for the good comments. Unlike common perovskite materials such as BNT (sodium bismuth titanate), the tungsten bronze structure is most stable in the tetragonal phase, especially for the unfilled SBN system. Only the transition from ferroelectric phase to paraelectric phase can be observed in the dielectric temperature spectrum, which is different from the ferroelastic-ferroelectric-paraelectric multiple phase transition in the “filled” $\text{Sr}_2\text{NaNb}_5\text{O}_{15}$ ceramics³². The absence of such transitions in the in-situ XRD results may be due to factors of experimental technique. The sensitivity and resolution of the XRD technique used in the experiment may not be sufficient to detect subtle phase transitions or structural changes in the material. Some TTB materials may exhibit high crystal structure stability over a wide range of temperatures, making phase transitions less likely or less pronounced. In addition, similar XRD patterns exhibiting excellent temperature insensitivity also appear in unfilled $\text{Sr}_{0.425}\text{La}_{0.1}\square_{0.05}\text{Ba}_{0.425}\text{Nb}_{1.4}\text{Ta}_{0.6}\text{O}_6$ ceramics³³, showcasing no detectable changes within the temperature range of -160 to 290 °C. In essence, the temperature-insensitive crystal structure constitutes one of the crucial factors contributing to the outstanding temperature stability observed in the energy storage performance of BSCN0.30 ceramics.

Comment 7. For practical applications, the evaluation of the discharge capability

of ceramic capacitors is very important. I suggest the authors to supplement relevant experimental results, including discharge speed and temperature stability. In addition, the detailed parameters of the discharge circuit should also be given, since the discharge behavior critically depends on the construction of the circuit.

Response 7. Thanks for the valuable comment. According to this comment, we have added the results of discharge capability and studied the temperature stability.

The details of the charge and discharge circuit are as follows: The ceramic sample has a thickness of 0.15 mm and a silver electrode area of 0.032 cm². A load resistor with a resistance of 300 Ω is utilized for overdamped charge and discharge measurements. The temperature change process is meticulously regulated, with the sample immersed in dimethyl silicone oil throughout the duration of the test.

Detailed modifications in the manuscript: (see page 25 in the revised manuscript)

In practical applications, assessing the charge and discharge capabilities holds paramount importance. To evaluate the real charge and discharge performance of BSCN0.30 ceramics, both under-damped and over-damped charge and discharge tests were conducted. Additionally, the alterations in charge and discharge performance at various temperatures were thoroughly evaluated. The results are presented in Figures S9 and S10. The findings reveal that BSCNT attains an impressive current density (C_D) of up to 1500 A/cm², a power density (P_D) of 280 MW/cm³, a discharge energy storage density (W_{diss}) of 2.6 J/cm³, and a discharge speed of $t_{0.9}=69.4$ ns. Notably, BSCNT0.30

showcases remarkable temperature stability, underscoring its immense potential for use in demanding operational environments, such as high-temperature settings.

Detailed modifications in the Supporting Information: (see pages 10 and 11 in the revised Supporting Information)

Figure S9. (a) underdamped charge-discharge curves of BSCNT0.30 ceramics at various electric fields. (b) C_D and P_D of BSCNT0.30 ceramics at various electric fields. (c) underdamped charge-discharge curves (@380 kV/cm) of BSCNT0.30 ceramics at various temperature. (d) C_D and P_D (@380 kV/cm) of BSCNT0.30 at various temperature.

Figure S10. (a) The overdamped charge-discharge as a function of time at various electric fields.

(b) W_{dis} as a function of time at various electric fields. (c) The variation of W_{dis} and $t_{0.9}$ with electric

field. (d) The overdamped charge-discharge as a function of time at various temperature. (e) W_{dis} as

a function of time at various temperature. (f) The variation of W_{dis} and $t_{0.9}$ with temperature.

References

1. Glass AM. Investigation of the Electrical Properties of $\text{Sr}_{1-x}\text{Ba}_x\text{Nb}_2\text{O}_6$ with Special Reference to Pyroelectric Detection. *J Appl Phys* **40**, 4699-4713 (1969).
2. Li C, Zhang Y, Liu J, Graetsch HA. Long-Range and Local Structure of $\text{Sr}_x\text{Ba}_{1-x}\text{Nb}_2\text{O}_6$ ($x = 0.33$ and 0.67) across the Ferroelectric–Relaxor Transition. *Chem Mater* **32**, 1844-1853 (2020).
3. Liu H, Dkhil B. Origin of the crossover from ferroelectric to relaxor in tetragonal tungsten bronzes. *J Alloys Compd* **929**, 167314 (2022).
4. Hai L, Liu H. Effect of A-site occupancy on phase transition and electric properties of $\text{Ca}_x\text{Sr}_{0.6-x}\text{Ba}_{0.4}\text{Nb}_2\text{O}_6$ ceramics. *Ceram Int* **49**, 22211-22218 (2023).
5. Hsu W-L, Tsai C-W, Yeh A-C, Yeh J-W. Clarifying the four core effects of high-entropy materials. *Nature Reviews Chemistry*, (2024).
6. Chen Z. Designs where disorder prevails. *Science* **384**, 158-159 (2024).
7. Cao L, Yuan Y, Li E, Zhang S. Relaxor regulation and improvement of energy storage properties of $\text{Sr}_2\text{NaNb}_5\text{O}_{15}$ -based tungsten bronze ceramics through B-site substitution. *Chem Eng J* **421**, 127846 (2021).

8. Zhao P, Tang B, Si F, Yang C, Li H, Zhang S. Novel Ca doped Sr_{0.7}Bi_{0.2}TiO₃ lead-free relaxor ferroelectrics with high energy density and efficiency. *J Eur Ceram Soc* **40**, 1938-1946 (2020).
9. Chen L, *et al.* Large Energy Capacitive High-Entropy Lead-Free Ferroelectrics. *Nano-Micro Letters* **15**, 65 (2023).
10. Colomban P, Romain F, Neiman A, Animitsa I. Double perovskites with oxygen structural vacancies: Raman spectra, conductivity and water uptake. *Solid State Ionics* **145**, 339-347 (2001).
11. Zhang L, Pu Y, Chen M. Complex impedance spectroscopy for capacitive energy-storage ceramics: a review and prospects. *Mater Today Chem* **28**, 101353 (2023).
12. Levin I, Stennett MC, Miles GC, Woodward DI, West AR, Reaney IM. Coupling between octahedral tilting and ferroelectric order in tetragonal tungsten bronze-structured dielectrics. *Appl Phys Lett* **89**, (2006).
13. Zhu X, *et al.* A Crystal-Chemical Framework for Relaxor versus Normal Ferroelectric Behavior in Tetragonal Tungsten Bronzes. *Chem Mater* **27**, 3250-3261 (2015).
14. Cao L, *et al.* Low temperature relaxor, polarization dynamics and energy storage properties of Ca_{0.28}Ba_{0.72}Nb₂O₆ tungsten bronze ceramics. *Chem Eng J* **479**, 147664 (2024).
15. Gao Y, *et al.* Ultrahigh Energy Storage in Tungsten Bronze Dielectric Ceramics Through a Weakly Coupled Relaxor Design. *Adv Mater* **36**, 2310559 (2024).
16. Stennett MC, *et al.* Dielectric and structural studies of Ba₂MTi₂Nb₃O₁₅ (BMTNO₁₅, M=Bi³⁺,La³⁺,Nd³⁺,Sm³⁺,Gd³⁺) tetragonal tungsten bronze-structured ceramics. *J Appl Phys* **101**, (2007).
17. Mao MM, Li K, Zhu XL, Chen XM. Incommensurate and commensurate modulations of Ba₅RTi₃Nb₇O₃₀ (R = La, Nd) tungsten bronzes and the ferroelectric domain structures. *J Appl Phys* **117**, (2015).
18. García-González E, Torres-Pardo A, Jiménez R, González-Calbet JM. Structural Singularities in Ferroelectric Sr₂NaNb₅O₁₅. *Chem Mater* **19**, 3575-3580 (2007).
19. Xu S, *et al.* Enhancing Energy Storage Performance in Lead-Free Bismuth Sodium Niobate-Based Tungsten Bronze Ceramics through Relaxor Tuning. *ACS Appl Mater Interfaces* **15**, 11642-11651 (2023).
20. Schneck J, Toledano JC, Whatmore R, Ainger FW. Incommensurate phases in ferroelectric tetragonal tungsten bronzes. *Ferroelectrics* **36**, 327-330 (1981).

21. Zhang YY, Chen L, Liu H, Deng SQ, Qi H, Chen J. High-performance ferroelectric based materials via high-entropy strategy: Design, properties, and mechanism. *Infomat* **5**, e12488 (2023).
22. Zhang M, *et al.* Ultrahigh energy storage in high-entropy ceramic capacitors with polymorphic relaxor phase. *Science* **384**, 185-189 (2024).
23. Li Q, *et al.* Flexible high-temperature dielectric materials from polymer nanocomposites. *Nature* **523**, 576-579 (2015).
24. Pan H, *et al.* Ultrahigh energy storage in superparaelectric relaxor ferroelectrics. *Science* **374**, 100-104 (2021).
25. Shvartsman VV, Lupascu DC. Lead-Free Relaxor Ferroelectrics. *J Am Ceram Soc* **95**, 1-26 (2012).
26. Shvartsman VV, Dec J, Miga S, Łukasiewicz T, Kleemann W. Ferroelectric Domains in $\text{Sr}_x\text{Ba}_{1-x}\text{Nb}_2\text{O}_6$ Single Crystals ($0.4 \leq x \leq 0.75$). *Ferroelectrics* **376**, 1-8 (2008).
27. Feng WB, Zhu XL, Liu XQ, Chen XM. Crystal structure, ferroelectricity and polar order in a $\text{Ba}_4\text{R}_2\text{Zr}_4\text{Nb}_6\text{O}_{30}$ (R = La, Nd, Sm) tetragonal tungsten bronze new system. *J Mater Chem C* **5**, 4009-4016 (2017).
28. Feng WB, *et al.* Relaxor nature in $\text{Ba}_5\text{RZr}_3\text{Nb}_7\text{O}_{30}$ (R = La, Nd, Sm) tetragonal tungsten bronze new system. *J Am Ceram Soc* **101**, 1623-1631 (2018).
29. Shannon RD. Revised effective ionic radii and systematic studies of interatomic distances in halides and chalcogenides. *Acta Crystallographica Section A* **32**, 751-767 (1976).
30. Yang ZJ, Liu XQ, Zhu XL, Chen XM. Crossover from normal to relaxor ferroelectric in $\text{Sr}_{0.25}\text{Ba}_{0.75}(\text{Nb}_{1-x}\text{Ta}_x)_2\text{O}_6$ ceramics with tungsten bronze structure. *Appl Phys Lett* **117**, (2020).
31. Ke S, Fan H, Huang H, Chan HLW, Yu S. Dielectric, ferroelectric properties, and grain growth of $\text{Ca}_x\text{Ba}_{1-x}\text{Nb}_2\text{O}_6$ ceramics with tungsten-bronzes structure. *J Appl Phys* **104**, (2008).
32. Li L, Yang B, Chao X, Wu D, Wei L, Yang Z. Effects of preparation method on the microstructure and electrical properties of tungsten bronze structure $\text{Sr}_2\text{NaNb}_5\text{O}_{15}$ ceramics. *Ceram Int* **45**, 558-565 (2019).
33. Peng H, *et al.* Superior Energy Density Achieved in Unfilled Tungsten Bronze Ferroelectrics via Multiscale Regulation Strategy. *Adv Sci (Weinh)* **10**, e2300227 (2023).

REVIEWERS' COMMENTS

Reviewer #1 (Remarks to the Author):

Thanks to the authors for considering all comments.
On my part there is nothing more to request.

Reviewer #3 (Remarks to the Author):

I have reviewed the author's revisions to all comments, including but not limited to my own. I believe the author has addressed my confusion and the scientific issues in the manuscript. The current manuscript proposes a new chemical composition design approach for TTB to improve its energy storage performance. I recommend the current version of the manuscript for publication in NC.

Response Letter to Reviewers

We appreciate the reviewers for their efforts and time spent on our manuscript. The comments are highly valuable and extremely helpful for us to further improve the quality of our manuscript.

Reviewer #1:

Thanks to the authors for considering all comments.

On my part there is nothing more to request.

Response: We are deeply grateful to the reviewer for reviewing our manuscript and agreeing to publish it.

Reviewer #3:

I have reviewed the author's revisions to all comments, including but not limited to my own. I believe the author has addressed my confusion and the scientific issues in the manuscript. The current manuscript proposes a new chemical composition design approach for TTB to improve its energy storage performance. I recommend the current version of the manuscript for publication in NC.

Response: We are deeply grateful to the reviewer for reviewing our manuscript and agreeing to publish it.